# A DEXiRE for Extracting Propositional Rules from Neural Networks via Binarization

Victor Contreras [1], Niccolo Marini [1], Lora Fanda [2], Gaetano Manzo [1,3], Yazan Mualla [4], Jean-Paul Calbimonte [1,5], Michael Schumacher [1] and Davide Calvaresi [1,*]

1    Institute of Information Systems (IIG), University of Applied Sciences and Arts Western Switzerland (HES-SO), 3960 Sierre, Switzerland
2    Department of Basic Neurosciences, Faculty of Medicine, University of Geneva, 1211 Geneva, Switzerland
3    National Institutes of Health (NIH), Baltimore, MD 21224, USA
4    CIAD UMR 7533, Univ. Bourgogne Franche-Comté, UTBM, F-90010 Belfort, France
5    The Sense Innovation & Research Center, 1007 Lausanne, Switzerland
*    Correspondence: davide.calvaresi@hevs.ch

**Abstract: Background:** Despite the advancement in eXplainable Artificial Intelligence, the explanations provided by model-agnostic predictors still call for improvements (i.e., lack of accurate descriptions of predictors' behaviors). **Contribution:** We present a tool for Deep Explanations and Rule Extraction (DEXiRE) to approximate rules for Deep Learning models with any number of hidden layers. **Methodology:** DEXiRE proposes the binarization of neural networks to induce Boolean functions in the hidden layers, generating as many intermediate rule sets. A rule set is inducted between the first hidden layer and the input layer. Finally, the complete rule set is obtained using inverse substitution on intermediate rule sets and first-layer rules. Statistical tests and satisfiability algorithms reduce the final rule set's size and complexity (filtering redundant, inconsistent, and non-frequent rules). DEXiRE has been tested in binary and multiclass classifications with six datasets having different structures and models. **Results:** The performance is consistent (in terms of accuracy, fidelity, and rule length) with respect to the state-of-the-art rule extractors (i.e., ECLAIRE). Moreover, compared with ECLAIRE, DEXiRE has generated shorter rules (i.e., up to 74% fewer terms) and has shortened the execution time (improving up to 197% in the best-case scenario). **Conclusions:** DEXiRE can be applied for binary and multiclass classification of deep learning predictors with any number of hidden layers. Moreover, DEXiRE can identify the activation pattern per class and use it to reduce the search space for rule extractors (pruning irrelevant/redundant neurons)—shorter rules and execution times with respect to ECLAIRE.

**Keywords:** eXplainable Artificial Intelligence (XAI); binary neural networks; rules extraction; rules induction; global explanations

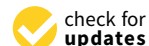



## 1. Introduction

Deep learning (DL) comprises a set of machine learning (ML) methods based on artificial neural networks (ANN) that can learn complex representations directly from input data through representation learning [1,2]. DL techniques are employed in complex tasks where manual feature engineering can be time-consuming or impossible to design. Moreover, DL architectures can be generalizable, and pre-trained DL models can be used across domains. For example, DL predictors have been adopted in numerous applications, including image processing [3], recommender systems [4], cyber-security [5], natural language processing [6], and in multiple decision support systems (DSS) [7]. DSS can strongly influence and persuade individuals and organizations. This raises ethical concerns about trust, privacy, and accuracy [8], as well as legal questions about liability, transparency, and risk exposition [9]. Therefore, DSS's inner workings of the principles/mechanisms and outcomes must be intelligible to accept the DSS conclusions.

Interpreting and explaining DL predictors (both from human [10] and virtual agents [11,12] perspectives) have been, to date, widely investigated in the discipline named eXplainable Artificial Intelligence (XAI) [13]. XAI methods can be categorized into *intrinsically interpretable* (interpretable by design—i.e., decision trees [14], linear regression [15], and rule sets [15,16]), and *post-hoc explainable* (predictors trained on inputs–outputs—i.e., LIME [17], SHAPLEY [18], ECLAIRE [19], and CIU [20]).

Despite the significant advances in XAI, the quality of the explanations still calls for improvements. For example, on the one hand, model-agnostic approaches do not provide highly accurate descriptions of the predictor's internal behavior. Instead, they focus solely on surrogate models (intrinsically interpretable models that mimic the original models) and ignore the internal structure of the actual model. On the other hand, model-specific approaches are restricted to a few DL architectures, limiting their scope of explanation. DL predictors are connectionist models that store their knowledge through a distributed structure represented by weighted connections between neurons and non-linear activation functions. Decompositional rule-extraction methods are suitable for extracting knowledge stored in the DL predictors. However, their scope is limited to specific architectures, usually one-hidden layer neural networks, and it is computationally expensive (requiring several phases of pruning and training).

This paper presents a tool for Deep Explanations and Rule Extraction (DEXiRE). Such a tool approximates rules for DL models with any number of hidden layers. The proposed approach employs binary neural networks to induce Boolean functions in the hidden layers, generating intermediate rule sets for every hidden layer. In turn, a rule set is inducted between the first hidden layer and the input layer. Finally, the complete rule set is obtained using inverse substitution on intermediate rule sets and first-layer rules. To reduce the size and complexity of the final rule set, the algorithm uses statistical tests and satisfiability algorithms to filter redundant, inconsistent, and non-frequent rules.

The rest of the paper is organized as follows:
Section 2 presents the state of the art on DL, XAI, and rule extraction from ANNs. Section 3 describes the methodology. Section 4 presents the experimental setup and the tests' execution. Section 5 presents the experimental results. Section 6 discusses and evaluates the results and what we can learn from them. Finally, Section 7 concludes the paper.

## 2. State of the Art

The XAI contributions can be classified into two main branches: Explainable-by-design and Post-hoc Explanations [21,22].

As shown in Figure 1, on the one hand, decision trees, linear models, and rule-based approaches are considered explainable-by-design, meaning that the output is derived from the set of rules or thresholds within them. On the other hand, deep learning models, such as ANN, belong to post-hoc explanations, meaning that the output requires a third-party approach to be explained. Post-hoc explanations can be divided into local explanations (e.g., features importance) and global explanations (e.g., rule extraction). The concept of global or local explanations refers to interpreting a model from a different perspective. Global explanations focus on the model's perspective, while local explanations focus on a particular data point or observation [21].

This paper focuses on rule-extraction approaches, which elicit the hidden knowledge embedded in DL predictors and help explain the ANN outcomes. Extracted rules can be used for hazard mitigation, traceability, system stability and fault tolerance, operational verification and validation, and more [23,24]. Andrews et al. [25] propose a multidimensional taxonomy for several rule-extraction methods, from shallow ANN architectures. They use the (i) decompositional approach: rule-extraction algorithms that work on the neuron level; (ii) pedagogical approach: rule-extraction algorithm regards the neural network as a black box; and (iii) eclectic approach: the combination of both decompositional and pedagogical approaches.

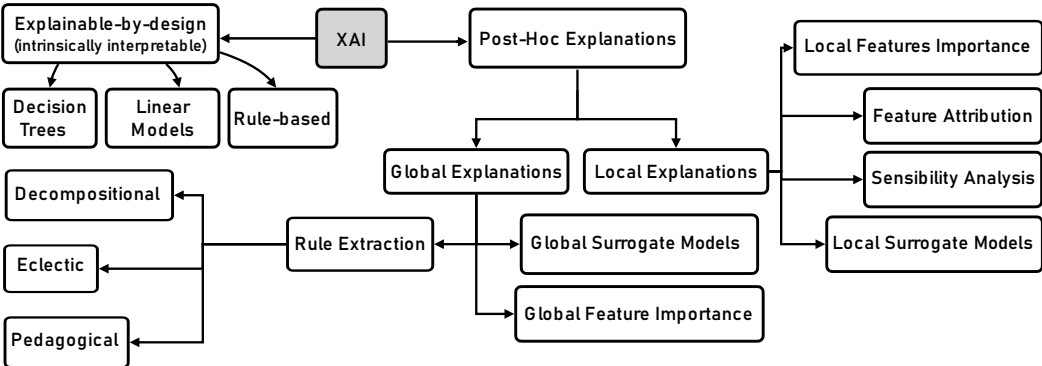

**Figure 1.** Taxonomy of the XAI classes of algorithms.

Decompositional algorithms work on the neuron level by extracting rules from neurons and inputting them into the whole network. In decompositional algorithms, every neuron and connection is transformed into a set of rules. A pioneer of decompositional algorithms is Fu [26,27], which proposed the KT algorithm and outlined the general algorithm by setting a threshold for each neuron. Fu resolved the problem by mapping the output result from each neuron (in hidden or output layers) into a Boolean function according to a given threshold. Towell and Shavlik [28] expanded the subset of Fu's algorithm, which can extract M-of-N rules from MLP models. However, a significant complexity of computing time was added for finding all potential sets of links per neuron unit. The authors in [29] extracted rules from a neural network with one single hidden layer and one linear output unit for regression problems. The activation function in the hidden layer is approximately by piece-wise linear functions. To reduce the complexity, the network is pruned before rule extraction. Despite this limitation, the inherent simplicity of decompositional algorithms makes them extremely useful for explaining the mechanics of rule extraction and for providing interpretability of ANN models at the level of individual hidden and output units.

*Pedagogical* rule-extraction algorithms consider ANN as a black box, extracting based on the ANN's inputs and outputs [30]—neurons and their connections are not subjected to analysis. The rules extracted using Pedagogical algorithms are directly extracted from the ANN's output, which changes as the ANN's input changes. Therefore, contrary to decompositional algorithms, the extracted rules are still intelligible if the ANN's internal structure is complex. Thrun in [31] extracted symbolic knowledge rules from ANN based on Validity Interval Analysis by propagating the activations through the network. This technique simulates the ANN train coupling the system's input and output patterns. HYPINV algorithm extracts rules from trained neural networks used in classification problems regardless of the structure of the network [32]. Model-agnostic methods, such as HYPINV, explain the predictions of arbitrary machine learning models independently of the implementation. They provide a way to explain predictors by considering models as black boxes. HYPINV approximates the network decision boundary by finding hyperplanes tangent to the decision hypersurface. Then the network decision boundary is represented in rules. Sethi et al. [30] proposed a pedagogical algorithm named KDRuleEx for rule extracting in the form of IF-THEN. The method can deal with discrete and continuous features and handles non-binary input. The authors in [33] compared rule sets extracted from ensembles of interpretable neural networks DIMLP, and their conclusion is that rule sets extracted from neural networks have greater accuracy than those extracted from decision tree ensembles but at a higher computational cost. The authors in [19] presented a new method to extract rules from Deep neural networks with any number of hidden layers. This efficient algorithm induces intermediate rule sets between the hidden layer and the predicted labels and with a process call-wise substitution to produce the final rule set.

*Eclectic* approach combines both the Decompositional and Pedagogical approaches. Most recent contributions in XAI are based on Eclectic approaches, such as Ebecken [34],

which present the RX algorithm to extract rules from trained MLP in classification problems. RX (i) applies a clustering genetic algorithm to find clusters of hidden unit activation values, and (ii) generates classification rules describing these clusters w.r.t. the input. Local Interpretable Model-agnostic Explanations (LIME) [35] is a technique that explains how the input characteristics of an ML model affect its predictions. For instance, for image classification tasks, LIME seeks to find the region of an image (set of super-pixels) having the strongest association with a prediction label. LIME creates explanations by generating a new dataset of random disturbances (with their respective predictions) around the explained instance. Then it fits a weighted local surrogate model. This local model is usually simpler with inherent interpretability, such as a linear regression model. The first theoretical analysis of LIME was published in [36], confirming the crucial role of LIME, but also demonstrating its limitation (i.e., poor choice of parameters causes LIME to miss essential features).

In [37], authors concluded that pruning algorithms based on Mutual Information and Significance could be more efficient than the methods based on sensitivity and magnitude, as they consider the mutual dependency between the inputs of the network and outputs of the hidden neurons. This is a comparative study of pruning algorithms that compares six different algorithms of large variability with details explanations and uses only one three-class problem application (out of four) since it is the Iris dataset (popular and generally used in creating/testing algorithms) that might suggest that the results should be looked at critically.

Local Interpretable Visual Explanations (LIVE) [38] uses a surrogate model to predict the black box model's local properties, generating the representative model's coefficients. LIVE differs from LIME regarding local discoverability and how it handles interpretable inputs. LIVE does not create an interpretable input space by transforming input features but instead uses the original feature space. Since the shuffled data points correspond very closely to the original points, the similarity between them is measured using the identity kernel, while the original features are used as an interpretable input.

SHapley Additive exPlanations (SHAP) [39] is a game theory-inspired method that tries to improve interpretability by calculating the importance of each feature in individual predictions. First, the author defines a class of feature attribution methods. It integrates six methods, including LIME [35], DeepLIFT [40], and Layer-Wise Relevance Propagation [41]. Then, they propose the SHAP values as unified measures of feature importance. These valued hold three desirable properties: local accuracy, missingness, and consistency. In the end, the authors present various methods for estimating SHAP values, demonstrating the superiority of these values in distinguishing various output classes, and better aligning with human intuitions compared to other methods [21].

The BreakDown method [40] is similar to SHAP [39], in trying to attribute the conditioned responses of a black-box model proportionally to the input features. However, unlike SHAP, the BreakDown method is greedy in terms of conditional responses as it considers only one series of nested conditional responses. This method is theoretically not as good as SHAP, but it is faster to compute and more natural to interpret.

Comparing LIVE, LIME, SHAP, and BreakDown, there is no best-for-all method developed to meet all requirements. Most methods are focused on a particular model or data type, or their scope is either local or global, but not both. Even though SHAP is nearly the most complete method, it is not without flaws; KernelSHAP, the kernel version of SHAP, like most permutation-based methods, does not consider feature dependencies, hence, emphasizing unlikely data points. TreeSHAP is the tree version of SHAP that could solve this problem. However, it relies on conditionally expected predictions that may give non-intuitive feature importance values, as features that have no impact on prediction could be given non-zero importance values [21].

The work in [42] shows that model trees can be used to build guidelines for learning problems, but this needs binary actions and does not apply to continuous action problems. Satisfiability (SAT) solvers are considered alternative methods, used mainly to construct decision trees and different constraint programming methods. SAT solvers such as [43,44]

and constraint programming methods like [45,46] work mainly on classification problems with binary input features and do not apply to robotics applications. Several SAT solvers and constraint programming solvers do not scale well to large datasets, mainly because these methods add per-sample constraints to the dataset [47].

### 3. DEXiRE: Rational, Methodology, Algorithm, and Performance Evaluation

This section presents a novel approach to extract rules from DL predictors (with any number of hidden layers) using binarization and Boolean rule induction. In particular, this study presents a rule-extraction approach to explain the inner behavior of DL classifiers—elaborating on the actual (and not a surrogated) model.

Approaching such a novelty, we have formulated (and validated) the following underlying hypotheses.

**Hypothesis 1.** *The binarization of hidden layers in DL predictors allows the identification of neuron activation patterns characteristic of each class.*

**Hypothesis 2.** *It is possible to use binary hidden layers to induce Boolean functions that approximate the behavior of each hidden layer.*

**Hypothesis 3.** *Intermediate rule sets approximating the behavior of each hidden layer can be combined to produce a final (global) rule set that describes the overall behavior of DL predictors.*

**Hypothesis 4.** *The verification of Hypotheses 1–3 implies the existence of rule sets able to explain binary and multiclass classifications of DL predictors.*

Figure 2 schematizes and highlights the relationship between the hypotheses, the evaluation metrics, and the algorithm's goal.

### 3.1. Underlying Rational Design

DEXiRE is a **decompositional rule-extraction algorithm** designed to test the Hypotheses 1–4 defined above. Overall, it combines activation analysis and **binary neural networks** to **extract logic rules** from deep neural networks (also referred to as rules induction). Figure 3 schematizes the working pipeline, and the listing Algorithm 1 details the internal steps in pseudocode.

**Decompositional rule extraction** methods extract rules at a neuron/layer level and combine them in the final rule set. Most of these methods are usually limited to networks with a single hidden layer and require pruning phases that involve retraining the model before rule extraction [19,48]. As a result, rule sets produced by these methods are longer and more difficult to interpret than those produced by pedagogical methods. DEXiRE uses satisfiability (SAT) algorithms [49–52], coverage [53], and information gain to reduce the size of intermediate and final rule sets without applying pruning phases.

**Binary neural networks (BNN)** are characterized by using binary weights and activations, reducing the memory space required to represent quantities. BNNs are suitable for working in limited-resource devices, improving the inference speed. However, in contraposition, BNNs may lose accuracy and precision, in particular when both activation and weights are binary [54–56]. DEXiRE employs binary neural networks to discretize only the neuron activations, based on Hypothesis 1, identifying active neurons, and making easier the induction of Boolean functions [57].

DL predictors are able to approximate real-value and Boolean functions. Choi et al. [58] and Mhaskar et al. [59] have shown that it is possible to compile feedforward neural networks into logic circuits, interpreting hidden layers as logic gates. Based on the previous works and Hypotheses 1 and 2, DEXiRE identifies the activation pattern per class in every hidden layer and then induces Boolean functions between the relevant neurons identified in the activation pattern.

**Rule induction** algorithms take a set of observations and, based on their statistical properties and correlations between features, identify rules that split the input space into

decision regions. Complex decision regions, such as those learned by DL predictors, produce complex rule sets that may be prone to overfitting. In order to avoid excessively complex rule sets, DEXiRE induces rule sets and prunes those rules with low coverage over the train samples. Once the intermediate rule sets are found, DEXiRE merges them into a final rule set, generated based on Hypotheses 3 and 4. Figure 2 conceptualizes the hypothesis (H1–H4) and their validation process through evaluation metrics (EM) w.r.t. DEXiRE's goal (G).

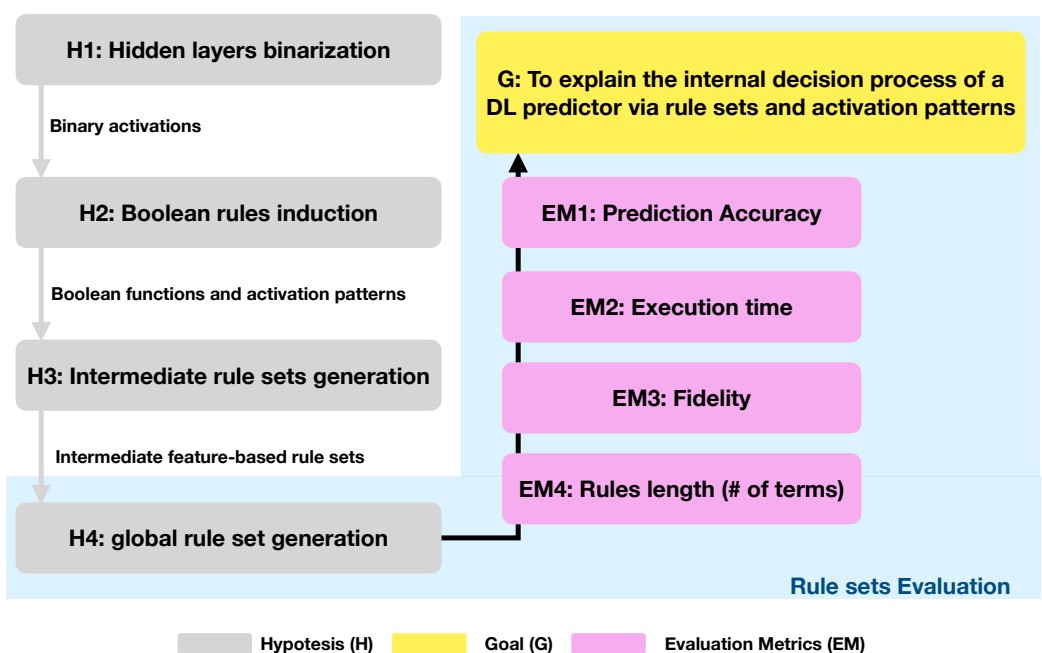

**Figure 2.** Overall conceptual schematization from Hypotheses (H) to the goal (G) via validation with Evaluation Metrics (EM).

### 3.2. DEXiRE Algorithm, Methodology, and Performance Evaluation

The following definitions are introduced to formally describe DEXiRE.

**Definition 1.** *An annotated training dataset is defined as the tuple* $< \mathbf{X}, \mathbf{y} >$*, where* $\mathbf{X} \in \mathbb{R}^{m \times n}$ *is a matrix with m training samples and n features and* $\mathbf{y} \in \mathbb{R}^{m \times l}$ *is the label matrix that assigns a label l to every sample in m.*

**Definition 2.** *A pretrained Deep Learning classification predictor is a function* $f_\Theta : \mathbf{X} \mapsto \hat{\mathbf{y}}$*, which maps the input features in* $\mathbf{X}$ *to the predicted labels* $\hat{\mathbf{y}}$*.*

**Definition 3.** *An inference rule is a logic function that draws a conclusion or conclusions by evaluating a premise. An inference rule can be expressed in an IF-THEN statement as follows:*

$$IF\ Premise\ THEN\ Conclusion \tag{1}$$

*Premises can be composed of different terms (logic expressions) linked by logic operators* ($\wedge$, $\vee$, $\neg$, $\oplus$)*. In this paper, rules are presented in IF-THEN form and premises are constructed from terms in disjunctive normal form (DNF).*

$$IF\ \underbrace{\overbrace{((X_i \geq v_i)}^{term} \vee \ldots (X_j < v_j)) \wedge \ldots ((X_k \leq v_k) \vee \ldots (X_m > v_m))}_{premise/antecedent}\ THEN\ \underbrace{\hat{y}}_{conclusion} \tag{2}$$

**Definition 4.** *Accuracy of a predictor is a measure of the quality of predictions, which compares the prediction of a model with the ground-truth labels, counting the fraction of correct predictions over the total number of predictions (Equation ([3](#))). Accuracy is measured using different metrics like accuracy-score, F1-score, precision, recall, and other performance metrics, evaluating the predictions against the ground-truth labels on supervised datasets [60,61].*

$$Accuracy = \frac{Number\ of\ correct\ predictions}{Total\ number\ of\ predictions} \tag{3}$$

**Definition 5.** *Fidelity is a metric that compares the predictions from the original black-box model ($\hat{\mathbf{y}}$) and the predictions from an interpretable model ($\hat{\mathbf{y}}_{rs}$), measuring how reliable the explanations are in reflecting the underlying model's behavior (Equation ([4](#))). Fidelity is measured in terms of accuracy, F1-score, and other similarity measures, using the predictions of the black-box model as the ground truth [60–62].*

$$Fidelity = Accuracy(\hat{\mathbf{y}}, \hat{\mathbf{y}}_{rs}) \tag{4}$$

**Definition 6.** *Coverage can be defined as the number of data instances that activate a rule over the set of total data instances that belong to the same class $< \mathbf{X_i}, y_i >$, where $\mathbf{X_i} \subset \mathbf{X}$ is a subset of instances on $\mathbf{X}$ that belongs to class $y_i$ [53]. Let $r_i$ be a rule whose conclusion is $y_i$, coverage for this rule can be defined as follows:*

$$coverage(r_i) = \frac{number\ of\ instances\ in\ \mathbf{X}_i\ that\ activate\ r_i}{number\ of\ instances\ in\ \mathbf{X}_i} \tag{5}$$

**Definition 7.** *Rule length is a measure of the number of terms (atomic Boolean expressions, e.g., $(x_i \geq v_i)$) in a rule set.*

**Definition 8.** *An activation pattern is the set of most frequent discretized activation values for each hidden layer that characterizes each class.*

　　**DEXiRE Formal goal**: Given a training set ($< \mathbf{X}, \mathbf{y} >$), as was described in Definition [1](#), and a pretrained DL classifier ($f_\Theta$) as was described in Definition [2](#); DEXiRE aims to induce a set of rules in IF-THEN form, as was described in Definition [3](#), which defines a logic model or theory $M$ that approximates the behavior of the DL predictor $f_\Theta$ such that $M \vDash \hat{y}_{rs}$, the logic model ($M$) semantically entails ($\vDash$) conclusions ($\hat{y}_{rs}$).

　　Figure [3](#) and Algorithm [1](#) describe the DEXiRE pipeline step by step. The steps (S) are numbered and described below. Such a tool requires a trained model $f_\Theta$ and the training set, $\mathbf{X} \in \mathbb{R}^{m \times n}$ with $m$ samples and $n$ features.

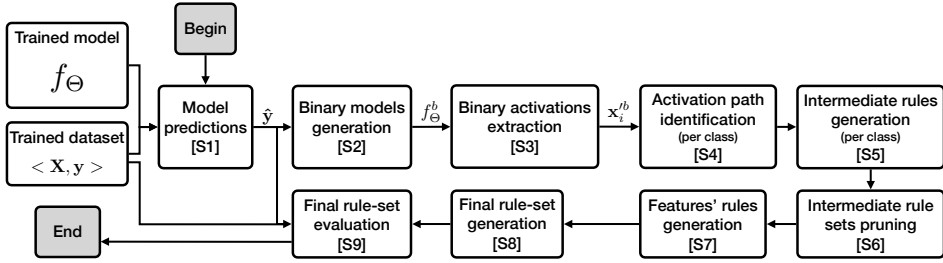

**Figure 3.** DEXiRE pipeline describes the series of steps executed by the DEXiRE algorithm (Algorithm [1](#)).

---

**Algorithm 1** DEXiRE algorithm pseudocode.

---

**Require:** Pretrained DL predictor $f_\Theta$ with $k+1$ hidden layers ($h_{0...k+1}$).
**Require:** Training feature matrix $\mathbf{X}$.
**Require:** Training labels matrix $\mathbf{y}$.
**Ensure:** Last layer on predictor $f_\Theta$ has as many neurons as classes.
1: $\hat{\mathbf{y}} \leftarrow f_\Theta(\mathbf{X})$                       ▷ Generates predictions from model $f_\Theta$ [S1]
2: $f_\Theta^b \leftarrow clone\_model(f_\Theta)$
3: **for** hidden layer i = 0,...k **do**
4:      $h_i^b \leftarrow binarize(h_i)$
5:      $f_\Theta^b \leftarrow replace\_layer(f_\Theta^b, h_i^b)$         ▷ Hidden layers are binarized producing a binary model [S2]
6: **end for**
7: $f_\Theta'^b \leftarrow fine\_tune(f_\Theta^b)$
8: **for** binary hidden layer i = 0,...k **do**
9:      $\mathbf{x}_i'^b \leftarrow h_i^b(\mathbf{x})$                    ▷ Binary patterns are extracted, [S3]
10: **end for**
11: **for** each class $c \in C$ **do**
12:      $activation\_pattern[c] \leftarrow \varnothing$
13:      **for** hidden layer i = 0,...k **do**
14:          $activation\_pattern[c][i] \leftarrow probability\_ranking(\mathbf{x}_i'^b)$     ▷ Activation pattern identification [S4]
15:      **end for**
16: **end for**
17: **for** each class $c \in C$ **do**
18:      $intermediate\_ruleset[c] \leftarrow \varnothing$
19:      $initial\_rule\_set[c] \leftarrow \varnothing$
20:      $final\_rule\_set[c] \leftarrow \varnothing$
21:      **for** hidden layer i = 0,...k **do**
22:          $intermediate\_ruleset[c][i] \leftarrow boolean\_inductor(activation\_pattern[c][i])$    ▷ Int. rule set generation [S5]
23:          $intermediate\_ruleset[c][i] \leftarrow prune(intermediate\_ruleset[c][i])$    ▷ Intermediate rule set running [S6]
24:      **end for**
25:      $feature\_rule\_set[c] = feature\_rule\_inductor(activation\_pattern[c][0])$    ▷ Feature rule set generation [S7]
26:      $final\_rule\_set[c] = rule\_merge(feature\_rule\_set[c], intermediate\_ruleset[c])$    ▷ Final rule set gen. [S8]
27: **end for**
28: $\hat{\mathbf{y}}_{rs} \leftarrow final\_rule\_set.predict(\mathbf{X})$
29: $accuracy \leftarrow accuracy\_function(\mathbf{y}, \hat{\mathbf{y}}_{rs})$              ▷ Final rule-set evaluation [S9]
30: $fidelity \leftarrow accuracy\_function(\hat{\mathbf{y}}, \hat{\mathbf{y}}_{rs})$

---

The DEXiRE pipeline (Algorithm 1) consists of several steps described as follows:

**S1:** **Model predictions**: Using the pre-trained DL predictor ($f_\Theta$) and the training set ($\mathbf{X}$), predicted labels ($\hat{\mathbf{y}}$) are obtained, as is described in line 1 on Algorithm 1 and in Equation (6).

$$\hat{\mathbf{y}} \leftarrow f_\Theta(\mathbf{X}) \tag{6}$$

**S2:** **Binary models generation**: To test Hypothesis 1, it is necessary to identify the most frequently active neurons per class. DEXiRE binarize the activation functions on hidden layers using the hard-tanh function, described in Equation (7) [63,64]. The hard-tanh activation function is commonly used in binary neural networks (BNN) to approximate the sign function, particularly during the backpropagation [65].
The binarized predictor $f_\Theta^b$ is created cloning (coping architecture and weights) from the original model $f_\Theta$, line 2 on Algorithm 1, then every hidden layer on $f_\Theta^b$ is replaced for a binary layer maintaining the same weights, lines 3–6 on Algorithm 1. Once the binary predictor $f_\Theta^b$ has been created, a fine-tuning training (line 7 in Algorithm 1) is executed on it to ensure high accuracy and fidelity.

$$\text{hard-tanh}(x) = \begin{cases} -1 \text{ if} & x < -1 \\ x \text{ if} & -1 \leq x \leq 1 \\ 1 \text{ if} & x > 1 \end{cases} \tag{7}$$

**S3:** **Binary activations extraction**: Binary activations are discrete approximations of logic values obtained from the hard-tanh activation function. Binary activation for hidden layer $i$ is defined as follows:

$$\mathbf{x}_i'^b : \mathbf{x}_i'^b \in \{1, -1\}^j \tag{8}$$

Binary neural network ($f_\Theta^b$) is used to obtain the binary activations $\mathbf{x}_i^{\prime b}$ for each data sample $x^{(r)}$ in the training set, $x^{(r)} \in \mathbf{X}$. The binary activation extraction process is executed for each binary hidden layer $\mathbf{h}_i^b$ (see Algorithm 1, lines 8–10).

**S4:** **activation pattern identification (per class)**: are obtained for each hidden binary layer $\mathbf{h}_i^{\prime b}$, in the previous step. Binary activations are grouped by class, and then the most frequent activation values are identified and stored in the dictionary (*activation_path*). Then a probability ranking is constructed per class using the likelihood of activation values per class, which describes the activation pattern per class as the collection of frequently activated neurons per layer (Algorithm 1 lines 11–15). Step S4 aims to test Hypothesis 1 identifying the activation pattern per class. Two example activation patterns are shown in Figures 4 and 5.

**S5:** **Intermediate rules generation (per class)**: Intermediate rules are generated, per class, from the activation pattern. Per each hidden layer and class, the most frequently activated neurons are selected, and a Boolean rule is inducted using the Sum of Products (SoP) or OneR methods. Intermediate rules are Boolean and operate with logic values, interpreting binary activation value 1 as the logic *True* and binary activation value $-1$ as the logic *False*. (1 lines 21–24). In this step, Hypothesis 2 is tested.

**S6:** **Intermediate rule sets pruning**: The intermediate rule sets are pruned based on the probability ranking (likelihood of activation) of the terms which compose the Boolean function and applying satisfiability (SAT) algorithms that check the consistency of the intermediate Boolean functions (Algorithm 1 line 23).

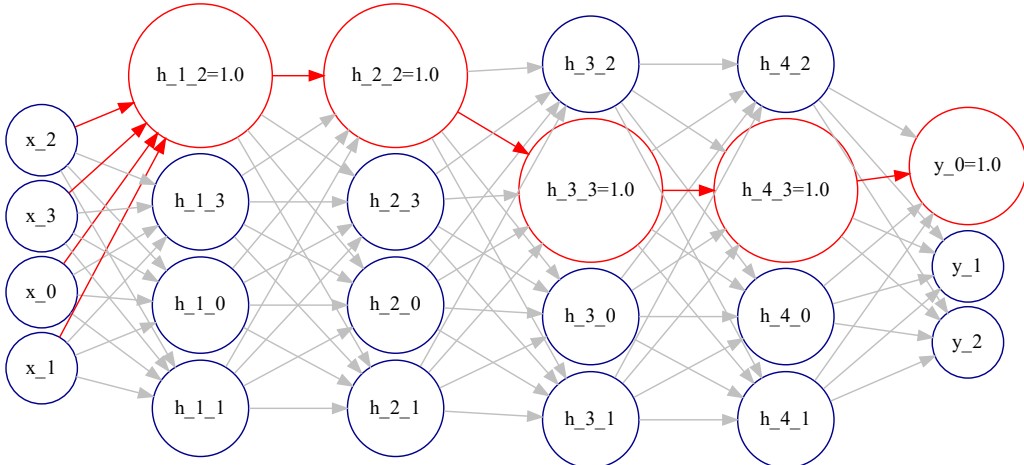

**Figure 4.** The activation pattern (the most frequently activated neurons) for class 0 is highlighted in red over the DL predictor architecture (Iris dataset). In addition, the most frequent activation values are shown for neurons in the activation pattern.

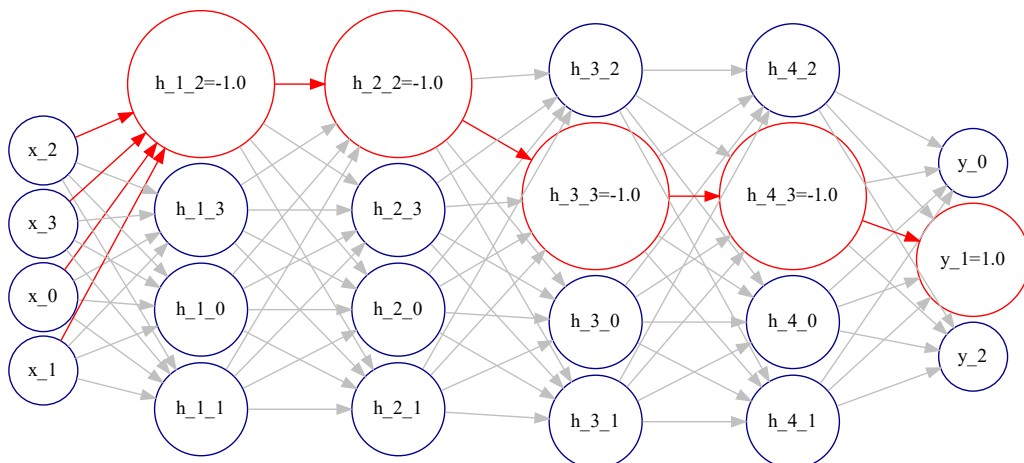

**Figure 5.** The activation pattern (the most frequently activated neurons) for class 1 is highlighted in red over the DL predictor architecture (Iris dataset). In addition, the most frequent activation values are shown for neurons in the activation pattern.

**S7:** **Features' rule generation**: Binary activation of the first hidden layer ($\mathbf{h}_0'^b$) is used as the target label to induce features' rule set.

Explainable layers (ExpL) are hidden layers that are able, through their weights and activation, to explain the underlying neuron behavior [60]. ExpL layers can learn an activation threshold $t_i$ for the features producing partitions in the feature space, in the form $(x_i \geq t_i)$, according to the class label. However, the partitions on the input space are binary and cannot be extended to the multiclass problem. For this reason, ExpL as a method for feature rule induction is only suitable for the binary classification task. For the multiclass task, we employ algorithms such as decision trees, random forest, ID3, CART, C4.5, and OneR. (Algorithm 1 line 25).

**S8:** **Final rule set generation**: Per class, intermediate rule sets are merged with the initial_rule_set to produce the final_rule_set (Algorithm 1 line 26). In this step, Hypotheses 3 and 4 are tested. The substitution process carried out to produce the final rule set can be visualized in Figure 6.

$$IF \; \neg(X_i < v_i) \; \wedge \; (X_j < v_j) \dots \; \wedge (X_k < v_k) \; THEN \; CLASS\_N$$

Final Rule set

$$IF \; (\neg A \wedge B \cdots \wedge Z) \, THEN \, CLASS\_N$$

Substitution  Substitution  Substitution

$$(X_i < v_i) \quad (X_j < v_j) \quad \dots \quad (X_k > v_k)$$

**Figure 6.** Substitution and merging process between the intermediate rule sets and the feature rule set employed to generate the final rule set.

**S9:** **Final rule-set evaluation**: Once the final rule set has been generated, its predictions $\hat{\mathbf{y}}_{rs}$ are evaluated against the true labels (**y**) predictions ($\hat{\mathbf{y}}$) to measure rule accuracy and fidelity (Algorithm 1 lines 28 and 30).

## 4. Experimental Setup

This section elaborates on the experimental setup, datasets used for the evaluation, and experimental process adopted to verify the hypotheses outlined in Section 3. Figure 7 shows the experimental workflow. A detailed description of each experimental step (ES) follows.

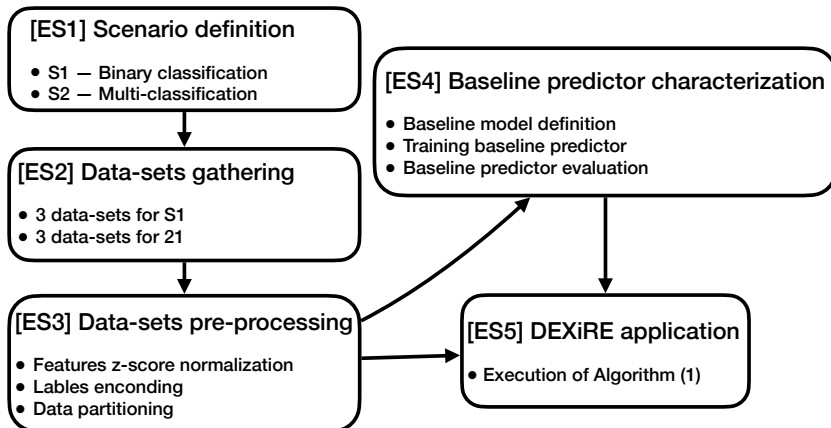

**Figure 7.** Experimental setup workflow describes the experimental steps (ES) carried out to test the DEXiRE algorithm.

### 4.1. ES1 Scenario Definition

Recalling that DEXiRE is a decompositional rule-extraction method designed to extract rules from DL classifiers, its features' rule generation (step S7 of the pipeline) operates differently for binary and multiclass classification. On the one hand, for binary classification (SC1), S7 consists of learning the initial rule set (the rule set between the input features and the first hidden layer) using explainable layers (ExpL). ExpL is an additional layer added to the original mode that can learn the activation threshold that splits the decision region between two classes [60]. On the other hand, multiclass classification (SC2) concerns three or more target categories. In the case of SC2, S7 employs algorithms like ID3, C4.5, and OneR to induce the initial rule set for each class.

### 4.2. ES2 Datasets Gathering

For this study, we selected publicly accessible datasets. Public datasets foster the replication and extension of our study. In particular, the datasets we employed are:

- **SC1 Binary classification datasets**:
  - **Breast Cancer Wisconsin (Diagnosis)**: The Breast Cancer Wisconsin Diagnosis (BCWD) dataset is a binary classification tabular dataset with 30 features and 569 instances. The BCWD dataset includes annotations to determine if a given data instance describes a malign (M) or benign (B) breast tumor [66]. A detailed description of features can be found in Appendix A.1.
  - **Banknote authentication Dataset** The banknote dataset is a binary classification tabular dataset with 1372 instances and four continuous features extracted from the instances' image wavelet transform. In addition, the banknote dataset includes annotations to classify a banknote as authentic or fake [67]. A detailed description of features can be found in Appendix A.2.
  - **Prima Indians diabetes mellitus classification** The Prima Indians diabetes dataset is a binary classification tabular dataset that contains 768 instances (patients) and eight features. The Prima Indians diabetes dataset includes annotations to determine if a given patient is diabetic or not [68]. A detailed description of features can be found in Appendix A.3.

- **SC2 Multiclass classification datasets**:
  - **Digits Dataset**: The Digits dataset is a multiclass classification tabular dataset with 1797 instances and 64 features, each one representing the intensity of a gray-scale pixel. The Digits dataset provides annotations to classify digits between 0 and 9 [67]. A detailed description of features can be found in Appendix A.4.

- **Iris**: The Iris dataset is a popular multiclass classification tabular dataset that contains 150 instances and four features (petal length, sepal length, petal width, and sepal width). The Iris dataset provides annotations to classify an instance between one of three target categories (Setosa, Virginica, Versicolor). A detailed description of features can be found in Appendix A.5.
- **Wine quality**: The wine quality dataset is a multiclass classification tabular dataset with 178 instances and 13 numeric features. The wine quality dataset provides annotations to classify wine into one of three classes [67]. A detailed description of features can be found in Appendix A.6.

### 4.3. ES3 Datasets Pre-Processing

Data normalization has been the first step in the pre-processing pipeline. Normalization changes the scale and range of data to improve the training process and reduce the bias induced by the feature ranges. In this study, all features have been normalized using the Z-score normalization. The one-hot encoding procedure transforms the target labels (**y**) into binary vectors.

Once the dataset has been normalized and the labels encoded, the dataset is separated into three sets: training set (60%), validation set (20%), and test set (20%) using stratified sampling and a random seed of 42.

### 4.4. ES4 Baseline Predictor Characterization

In ES4, the baseline DL predictor is defined. The baseline DL predictor is a feedforward DL model, and its architecture is defined for each dataset based on the number and type of features and labels. In turn, it is trained using the training and validation sets as described in experimental step ES3 (Section 4.3). The trained baseline predictor is evaluated using the test set and the performance metrics on the task in order to check the generalization ability and establish the baseline measure. The performance metrics used in this study are accuracy-score for binary classification and F1-score with micro-average for the multiclass task.

### 4.5. ES5 DEXiRE Application

In ES5, DEXiRE is applied to extract rules from the DL predictor (see Algorithm 1, lines 11 to 28). In turn, the obtained final rule set is evaluated with the test set and predictions from the DL predictor (see Algorithm 1 lines 28 to 30), employing accuracy-score for scenario S1 and F1-score with micro-average in scenario S2. All experiments were executed on a laptop with Intel Core i7, 32 GB of RAM, and Windows 10. The software requirements to replicate the study are python (v3.7), TensorFlow (v2.3.0), Larq (v0.12.2).

## 5. Results

In this section, experimental results are presented, discriminated by scenario (scenario SC1 and scenario SC2), as was described in Section 4.1.

### 5.1. SC1 Binary Classification

Results per dataset are shown in Table 1. Results are obtained by applying the experimental setup described in Section 4 to three publicly available datasets. The initial model to be explained is a **DL predictor** ($F_\Theta$) without binarization, and it is used as input for both rule-extraction algorithms ECLAIRE and DEXiRE. The initial DL predictor ($F_\Theta$) was trained and tested using five-fold cross-validation. The accuracy result reported for each DL predictor without binarization is the average accuracy-score measured on each fold.

Scenario (SC1) was tested on three publicly available datasets. Datasets employed in this scenario were briefly described in Section 4.2. The first column in Table 1 identifies the dataset. For each dataset, the second column describes the approach employed to extract rules from the DL predictor ($f_\Theta$). In this scenario, two rule-extraction algorithms were tested, ECLAIRE and DEXiRE. ECLAIRE is the current state-of-the-art algorithm for

rule extraction [19] and the baseline method for rule extraction. DEXiRE is our proposed approach for rule extraction, described in further detail in Section 3.

The third column presents the average accuracy-score for ECLAIRE and DEXiRE on the test set. The fourth column describes the fidelity measure, which quantifies the similarity between the DL model's predictions and those from the extracted rule sets (Definition 5). The fifth column describes the average rule length (the number of terms on each rule set, Definition 7) resulting from five-fold cross-validation. Finally, the sixth column presents the average and standard deviation of the rule-extraction algorithms' execution time, calculated based on the time measured on 100 executions using the same model and train set.

The Breast Cancer Wisconsin Diagnosis (BCWD) dataset is the first dataset shown in Table 1. BCWD dataset consists of 30 features and two target classes (for a detailed description of this dataset, see Appendix A.1). The DL predictor's accuracy is 97% with a standard deviation of ($\approx$1%), which shows a consistent performance of this model. The rule sets generated by ECLAIRE have an average accuracy of 93% with a standard deviation of ($\approx$0.4%), whereas the rule sets generated by DEXiRE have an average accuracy of 89% with a standard deviation of ($\approx$1.3%). Example rule sets extracted from the DL predictor are presented in Appendix B.1. ECLAIRE's example rule set is shown in Table A9 and consists of four terms. Features employed in this rule set were *concave_points_worst* and *perimeter_worst*. Similarly, the example rule set produced by DEXiRE (Table A8) consists of four terms.

**Table 1.** Summary table Scenario SC1 (binary classification) showing results for rule-extraction approach, accuracy, fidelity, rule length, and execution time for each dataset. Numerical results are reported with *average value $\pm$ standard deviation*. The best results in each dataset are shown in bold.

| Dataset | Approach | Accuracy | Fidelity | Rule Length | Execution Time |
|---------|----------|----------|----------|-------------|----------------|
| BCWD | DL predictor | $0.97 \pm 0.010$ | NA | NA | NA |
| | ECLAIRE | **$0.93 \pm 0.004$** | **$0.93 \pm 0.015$** | $6.4 \pm 3.2$ | $471$ ms $\pm 18.3$ ms |
| | DEXiRE | $0.89 \pm 0.013$ | $0.90 \pm 0.013$ | **$4.8 \pm 2.0$** | **$178$ ms $\pm 1.2$ ms** |
| Banknote | DL predictor | $0.98 \pm 0.008$ | NA | NA | NA |
| | ECLAIRE | $0.94 \pm 0.036$ | $0.95 \pm 0.026$ | $13.8 \pm 0.4$ | $213$ ms $\pm 3.08$ ms |
| | DEXiRE | **$0.94 \pm 0.032$** | **$0.95 \pm 0.026$** | $13.8 \pm 0.4$ | **$212$ms $\pm 2.17$ ms** |
| Prima diabetes | DL predictor | $0.76 \pm 0.017$ | NA | NA | NA |
| | ECLAIRE | $0.73 \pm 0.025$ | $0.86 \pm 0.031$ | $11.8 \pm 5.1$ | $381$ ms $\pm 7.56$ ms |
| | DEXiRE | **$0.75 \pm 0.027$** | **$0.90 \pm 0.018$** | **$8.4 \pm 1.9$** | **$232$ms $\pm 3.37$ ms** |

The second dataset shown in Table 1 is the Banknote authentication dataset. The banknote dataset consists of four features, taken from wavelet transform applied to banknote images, and two target classes to decide whether a banknote is authentic. A detailed description of this dataset can be found in Appendix A.2. For this dataset, the DL predictor's accuracy is 98% with a standard deviation of 0.08%. The accuracy measure for ECLAIRE's rule set was, on average 94% with a standard deviation of 3.6%. Similarly, DEXiRE's rule set presents an average accuracy of 94% and a standard deviation of 3.2%. The accuracy difference between the DL predictor and the rule sets varies $\approx$4%. Meanwhile, ECLAIRE's and DEXiRE's rule sets have a fidelity score of 95% with a standard deviation of 2.6%. Execution time is comparable in both datasets. Examples of the extracted rule sets from the DL predictor are presented in Appendix B.2. An example of ECLAIRE's rule set for the banknote dataset is shown in Table A12. An example of DEXiRE's rule set is shown in Table A11. ECLAIRE's rule set example consists of 10 terms, whereas DEXiRE's consists of 14 terms. Both rule sets exclude the feature *entropy_of_image*.

The Prima Indian Diabetes dataset results are shown at the bottom of Table 1. The Prima diabetes dataset consists of eight features obtained from female patients and two target classes to classify a patient as diabetic or non-diabetic. A detailed description of this dataset can be found in Appendix A.3. The DL predictor's accuracy is 76% with a standard deviation of 1.7%. ECLAIRE's rule set has an average accuracy of 73% with a

standard deviation of 2.5%, and DEXiRE's rule set reports an average accuracy of 75% with a standard deviation of 2.7%. The accuracy variation between the baseline and the rule sets is ≈3%. ECLAIRE's rule set has a fidelity score of 86% with a standard deviation of 3.1%. DEXiRE's rule set has a fidelity score of 90% with a standard deviation of 1.8%. Example rule sets extracted from the DL predictor are presented in Appendix B.3. An example of ECLAIRE's rule set is shown in Table A15 consists of 22 terms, whereas an example of DEXiRE's rule set is shown in Table A14 and consists of 10 terms. DEXiRE's rule set example employs features: BMI, glucose, and pregnancies, whereas ECLAIRE's rule set uses the same features with the addition of *SkinThickness*.

### 5.2. SC2 Multiclass Classification

Scenario (SC2) is devoted to a multiclass task and was tested on three datasets. Scenario SC2 differs from scenario SC1 in the technique employed to induce the features' rule sets. While SC1 uses ExpL, SC2 employs a decision tree as a rule inductor. Datasets tested in this scenario are briefly described in Section 4.2. The first column in Table 2 identifies the dataset. For each dataset, the second column on Table 2 lists the approach employed to extract rules from the baseline predictor ($f_\Theta$). The third column presents the average accuracy for each approach measured on the test set, calculated as the average value of five-fold cross-validation training. The fourth column describes the fidelity measure (Definition 5), which quantifies the similarity between the DL model's predictions and those from the extracted rule sets. The fifth column describes the average rule length (the number of terms on each rule set, Definition 7) calculated as the average rule length in the rule set per fold in five-fold cross-validation. Finally, the last column presents the average and standard deviation of the rule-extraction algorithms' execution time, calculated as the average time of 100 executions employing a fixed DL predictor and train set.

The Digits dataset shown at the top of Table 2, which consists of 64 features and ten target classes (digits from 0 to 9). A detailed description of this dataset can be found in Appendix A.4). The DL predictor's accuracy is 96% with a standard deviation of 1%. The rule sets extracted with ECLAIRE have an accuracy of 82% with a standard deviation of 1%. The *coverage threshold* (Th) is a parameter that filters out those terms with low coverage levels reducing the rule sets' length. To evaluate the impact of *coverage threshold* (Th) on the accuracy, fidelity, rule length, and execution time, DEXiRE was tested on different coverage threshold levels. The first row in Table 2 shows DEXiRE results for different Th (60%, 50%, 25%) levels. The default Th value for DEXiRE is 60%; for this Th value, DEXIRE's rule sets obtained an average accuracy of 44% with a standard deviation of 1.49%. For a Th value of 50%, the average accuracy obtained is 50% with a standard deviation of 1.63%. Finally, for a Th value of 25%, the average accuracy obtained is 79%, with a standard deviation of 2.5%. Fidelity follows a similar behavior. For the Th value of 60%, the obtained average fidelity of 43% with a standard deviation of 1.47%. For the Th value of 50%, the average fidelity value is 50% with a standard deviation of 1.67%, and finally, for the Th value of 25%, the obtained average fidelity is 79% with a standard deviation of 2%.

The Iris dataset consists of four features, taken from flowers' petals and sepals width and length, with three target classes (Setosa, Virginica, and Versicolor) (Appendix A.5). Performance measures for the Iris dataset are shown in the middle of Table 2. The DL predictor's accuracy is 92% with a standard deviation of 5.7%. The average accuracy for ECLAIRE's rule sets is 89% with a standard deviation of 8.3%, while DEXiRE's rule sets obtained an average accuracy value of 90% with a standard deviation of 5.9%. ECLAIRE's rule set has obtained an average fidelity value of 89% with a standard deviation of 3.9%. For DEXiRE, the rule sets have obtained an average fidelity value of 90% with a standard deviation of 5.9%. Rule sets examples extracted from the DL predictor are presented in Appendix B.5. An example of ECLAIRE's rule sets is shown in Table A19 and consists of six terms, whereas an example DEXiRE's rule set is shown in Table A18 consists of four terms. ECLAIRE's rule set example includes *sepal_width* and *petal_width* whereas DEXiRE's rule set only employs pedal features (*petal_width* and *petal_lenght*).

The Wine quality dataset is the third last dataset shown at the bottom of Table 2. The Wine quality dataset comprises 13 features that describe factors affecting the wine's quality and taste. The target category is one of three classes of wine, a detailed description of this dataset can be found in Appendix A.6. The DL predictor's accuracy is 99% with a standard deviation of 1.4%. The average accuracy value for ECLAIRE's rule sets is 83% with a standard deviation of 7.3%. The DEXiRE's rule sets have obtained an average accuracy value of 87% with a standard deviation of 2.8%. The accuracy variation between the DL predictor and the rule sets is ≈17%. ECLAIRE's rule sets have an average fidelity value of 84% with a standard deviation of 8.4%. DEXiRE's rule sets have an average fidelity value of 87% with a standard deviation of 2.8%. Examples of rule sets extracted from the DL predictor are presented in Appendix B.6. An example of ECLAIRE's rule set is shown in Table A21 consists of six terms, whereas DEXiRE's rule set example is shown in Table A20, and consists of four terms. Both rule sets include the feature *proline*, *color_intensity*, and *magnesium features*.

**Table 2.** Summary table Scenario SC2 (multiclass classification) showing results for Rule extraction approach, Accuracy, fidelity, rule length, and execution time for each dataset. Numerical results are reported with *average value ± standard deviation*. The best results in each dataset are shown in bold.

| Dataset | Approach | Accuracy | Fidelity | Rule Length | Execution Time |
|---|---|---|---|---|---|
| Digits | DL predictor | $0.96 \pm 0.010$ | NA | NA | NA |
| | ECLAIRE | $\mathbf{0.82 \pm 0.010}$ | $\mathbf{0.81 \pm 0.010}$ | $81.0 \pm 8.1$ | $9200 \text{ ms} \pm 124 \text{ ms}$ |
| | DEXiRE (Th = 60%) | $0.44 \pm 0.149$ | $0.43 \pm 0.147$ | $\mathbf{37.6 \pm 3.0}$ | $\mathbf{442 \text{ ms} \pm 5.3 \text{ ms}}$ |
| | DEXiRE (Th = 50%) | $0.50 \pm 0.163$ | $0.50 \pm 0.167$ | $39.0 \pm 2.6$ | $481 \text{ ms} \pm 8.74 \text{ ms}$ |
| | DEXiRE (Th = 25%) | $0.79 \pm 0.025$ | $0.79 \pm 0.020$ | $44.8 \pm 1.0$ | $490 \text{ ms} \pm 6.94 \text{ ms}$ |
| Iris | DL predictor | $0.93 \pm 0.057$ | NA | NA | NA |
| | ECLAIRE | $0.89 \pm 0.083$ | $0.89 \pm 0.039$ | $5.6 \pm 1.0$ | $177 \text{ ms} \pm 2.19 \text{ ms}$ |
| | DEXiRE | $\mathbf{0.90 \pm 0.059}$ | $\mathbf{0.95 \pm 0.026}$ | $\mathbf{4.0 \pm 0.0}$ | $\mathbf{0.916 \text{ ms} \pm 0.002 \text{ ms}}$ |
| Wine quality | DL predictor | $0.99 \pm 0.014$ | NA | NA | NA |
| | ECLAIRE | $0.83 \pm 0.073$ | $0.84 \pm 0.084$ | $\mathbf{3.2 \pm 0.4}$ | $249 \text{ ms} \pm 5.84 \text{ ms}$ |
| | DEXiRE | $\mathbf{0.87 \pm 0.028}$ | $\mathbf{0.87 \pm 0.028}$ | $4.0 \pm 0.0$ | $\mathbf{209 \text{ ms} \pm 1.05 \text{ ms}}$ |

## 6. Analysis and Discussion

This section elaborates on the obtained results and discusses the points characterizing DEXiRE and its results.

### 6.1. Analysis Scenario (SC1)

It is worth recalling that the length of the rules refers to the number of Boolean terms composing a rule set (see Definition 7). The average rule length obtained by DEXiRE in scenario SC1 is reported in Table 1 column 4. Looking at it closely, it is possible to notice that for BCWD and Prima Indians diabetes datasets, the average rule length for DEXiRE's rule sets is 4.8 and 8.4, respectively. Both results are shorter than those obtained by ECLAIRE (6.4 and 11.8, respectively). The DEXiRE and ECLAIRE's average rule lengths for the banknote dataset are mostly similar, with a value of 13.8.

On the one hand, rule length overlaps can be due to the influence of the dataset characteristics and model internal structure (see Section 6.3.1). On the other hand, the difference (if any) in the average rule length relies on the DEXiRE's use of binarized hidden layers, explainable layers, and related activation patterns. Differently, ECLAIRE uses continuous activations and decision trees to induce the features and intermediate rule sets. DEXiRE and ECLAIRE reach the same objective (generate a rule from a DL predictor) with different approaches—yet obtain comparable performance. For example, from an accuracy/fidelity point of view, DEXiRE sensibly outperforms ECLAIRE on the Prima Indians diabetes dataset, it equates ECLAIRE w.r.t. in the banknote dataset, and it sensibly underperforms w.r.t. BCWD (see Table 1—columns 3 and 4). However, concerning the execution time, DEXiRE outperforms ECLAIRE (90% faster in the best-case scenario—BCWD, 48% w.r.t. the Prima Indianas dataset) and mostly identical w.r.t. the banknote dataset.

Although the results obtained by DEXiRE and ECLAIRE are comparable, DEXiRE is able to provide additional insights about the internal decision process carried out by the neurons in the DL predictor (i.e., identify those neurons and their output values highly related with the predicted output). This benefit is provided due to the binarization process and activation pattern generated per class within DEXiRE.

### 6.2. Analysis Scenario (SC2)

Table 2 column 4 shows the average rule length obtained by DEXiRE in the scenario SC2.

It is worth recalling that DEXiRE employs a *coverage threshold* that allows to control the number of terms in each rule set (intermediate and final) by setting a minimum coverage threshold (Th) percentage and filtering all the rules with coverage (Definition 6) below the threshold. In the digits dataset, the rule length is 73% shorter than the one produced by ECLAIRE. However, the accuracy and fidelity obtained by DEXiRE are very poor. To achieve the performance obtained by ECLAIRE, we have decreased the Th value. In particular, with a Th value of 25%, accuracy and fidelity are 3% and 2% away from matching the ECLAIRE's performance while improving on the average rule length, which is 57% shorter than the for ECLAIRE.

In the case of the Iris dataset, the average rule length obtained by DEXiRE is 33% shorter than the one obtained by ECLAIRE. Whereas in the case of the wine quality dataset, the ECLARE's average rule length is sensibly shorter than the one from DEXiRE. The difference (if any) in the average rule length depends on two main factors: (i) the interrelation between the dataset characteristics and the DL predictor's internal structure and (ii) the differences in rule-extraction algorithms' approaches. While DEXiRE employs binarized hidden layers and related activation patterns to extract rules, ECLAIRE uses continuous activations and decision trees to induce the features and intermediate and final rule sets.

Another benefit to DEXiRE's approach is SEEN in the overall execution time. DEXiRE outperforms ECLAIRE in all datasets (197% faster in the best-case scenario – Iris dataset, 17% w.r.t the wine quality dataset). Additionally, in this scenario, the standard deviation of the execution time is low for DEXIRE (never exceeding 9 ms). Low variance levels are due to the use of the activation pattern to identify the most relevant neurons in each hidden layer, thus reducing the time for the generation of the intermediate datasets.

In this scenario, DEXiRE and ECLAIRE exhibit comparable results in terms of accuracy and fidelity. While DEXiRE indicates a tendency to produce shorter rule sets and execution times due to the binarization process and activation pattern generated per class, DEXiRE is able to provide additional insights about the internal decision process carried out by the neurons in the DL predictor and filter out those neurons that are not relevant to the decision process carried out by the DL predictor.

### 6.3. Discussion

This section discusses the results obtained w.r.t. the Hypotheses 1–4 initially formulated.

#### 6.3.1. Data and Models Interdependency

DEXiRE can have similar rule sets' lengths with (partial) overlapping terms with ECLAIRE. Such overlap(s) can occur since, although they use different procedures, both rely on the same inputs (model and the training dataset) to induce the set of rules.

The final rule set(s) are logical formulations containing (the most relevant) data features integrating the intermediate rule sets. Both ECLAIRE and DEXiRE have similar procedures to induce (predict) intermediate rule sets, relying on the data features. DEXiRE can employ one of two methods (ExpL or decision trees) to generate the feature rule set (Step S7 in Section 3). Similarly, ECLAIRE employs decision trees to induce the features' rule set.

To understand the effect of the input features on the final rule set, let us analyze the case of the BCWD dataset. This dataset has 30 input features. Although the final rule sets generated by ECLAIRE and DEXiRE differ in the number of terms and the composition of

rules, most of them contain the feature *concave_points_worst*. This tendency raises a question: why is the feature *concave_points_worst* present in the majority of rule sets produced by different rule-extraction algorithms?

To answer this question, we must analyze the behavior of the feature *concave_points_worst* w.r.t the DL model's preconditions and the other features in the dataset. Figure 8 shows the feature ranking according to variance per class w.r.t. DL predictor's labels ($\hat{y}$). The features in the upper area present lower variance per class and higher *information gain* value (see Tables A8 and A9). Thus, there is a high probability that this feature is selected by the rule generation algorithms (ExpL, decision trees) and included in the final rule set.

Finally, we can infer that features with a high discrimination capability have high chances (yet no certainty) of being included in the final rule set, regardless of the rule induction algorithm employed—explaining the (partial) overlapping obtained in some datasets.

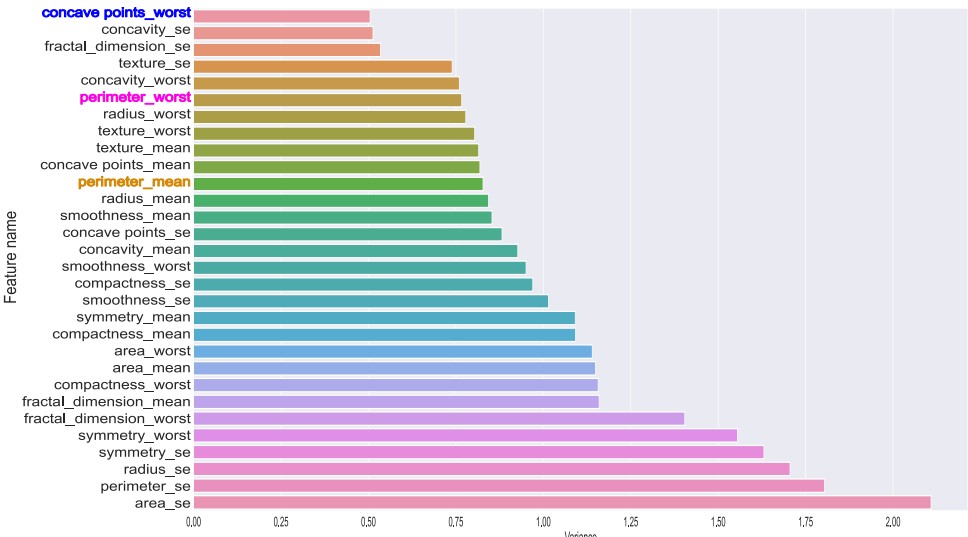

**Figure 8.** Feature ranking based on variance reduction per class w.r.t DL predictor's labels ($\hat{y}$). Features on the top of the figure present the highest variance reduction per class. Highlighted in bold blue is the feature *concave_point_worst* overlapping term between DEXiRE's and ECLAIRE rule sets. Highlighted in bold magenta is the *perimeter_worst* feature present in ECLAIRE's rule set, and highlighted in bold orange is the *perimeter_mean* feature present in DEXiRE's rule set.

Concerning the influence of the DL predictor's internal structure (architecture, activations, weights) on the final rule set, it is worth noticing that both DEXiRE and ECLAIRE are decompositional rule-extraction algorithms. This implies that they use the DL predictor's internal structure to induce rules and identify the neurons responsible for the decisions (verifying Hypotheses 1 and 2).

In conclusion, DL predictors and datasets are essential inputs for rule-extraction algorithms. For this reason, their composition and structure profoundly influence the final rule sets. Finally, it is worth highlighting the interdependence between the data and the DL predictors: the latter (being data-driven models) learn from the former, adjusting their structure and parameters to fit the task.

### 6.3.2. The Role of Binarization

Extracting rules from hidden layers in neural networks is still an open challenge. The hidden layer's activations are continuous and non-linear, which is an excellent characteristic for learning parameters with the back-propagation algorithm. However, it is more difficult to interpret their output values w.r.t. to the inputs. To solve this challenge, several decompositional rule-extraction algorithms approximate continuous activations through discrete segments to induce rules on each segment [29]. Nevertheless, these approaches have some limitations. For example, they tend to generate long rule sets (with many

terms/rules—not human-readable) and require several phases of pruning and re-training. Moreover, most decompositional approaches are limited to one or a predefined number of hidden layers. DEXiRE overcomes these limitations and simplifies the identification of frequent activation patterns (verifying Hypotheses 1–3).

In particular, Hypothesis 1 states that it is possible to identify activation patterns more simply by binarizing the hidden layers of a DL predictor ($f_\Theta$). To test Hypothesis 1, we binarized a trained DL predictor and then extracted the binary activations per hidden layer for each instance in the training set. Finally, using the binary activations, we calculate the kernel density estimation (KDE) for each neuron and visualize it in Figures 9 and 10.

Figure 9 shows the KDE for the first hidden binary layer (hidden layer 0) discriminated by class (class 0 and 1). Moreover, we can observe the high level of polarization present in neurons h_0_0, h_0_7, and h_0_8. Empty plots in this figure indicate neurons with zero variance (the same value for all the instances). Neurons with zero variance do not provide discriminative information, while highly polarized neurons become effective discriminators. Thus, the polarization induced by binary hidden layers and binary neural networks (BNN) allows the identification of activation patterns as described in Algorithm 1.

Figure 10 shows the KDE for the second hidden binary layer (hidden layer 1) discriminated by class (class 0 and 1). Similar to the previous layer, we can identify the relevant neurons in the decision of the classifier (neuron h_1_0) and discard those with zero variance. It is worth noticing that the number of discriminative neurons decreases, and the polarization increases as we approach the output layer. Thus, we can consider Hypotheses 2 and 3 validated. Additional KDE analysis for different datasets and DL predictors can be found in Appendix C.

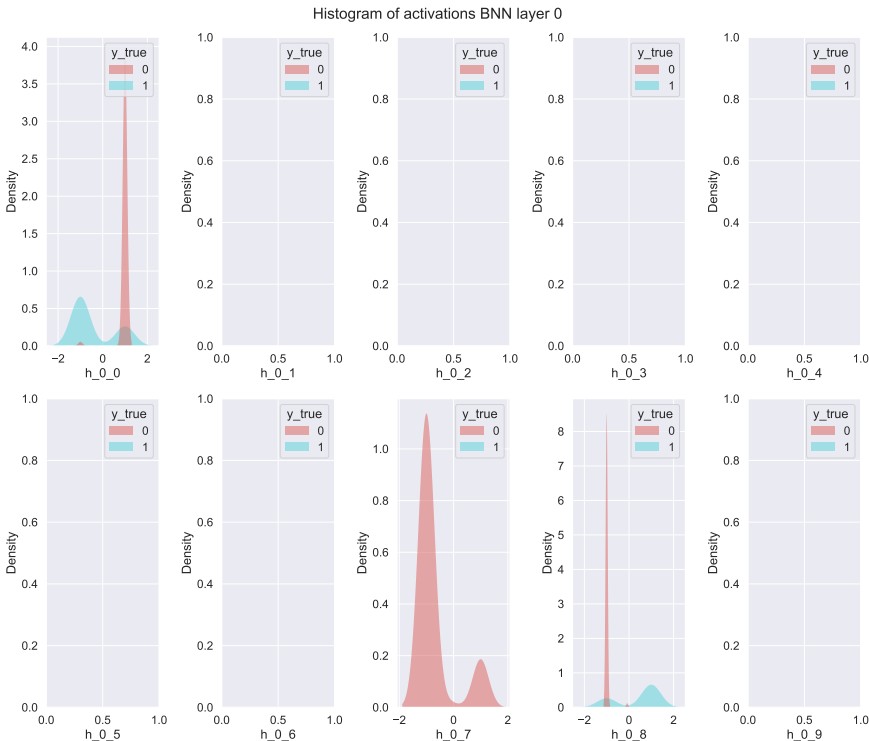

**Figure 9.** Histogram of binary activations generated employing KDE for neurons in hidden layer zero in DL predictor for BCWD dataset. Each plot shows the activation density function per neuron discriminated by class (class 0 is shown in red, and class 1 is shown in blue). Empty plots correspond to neurons that have zero variance and present the same activation value for all the samples, thus are not class-discriminatory.

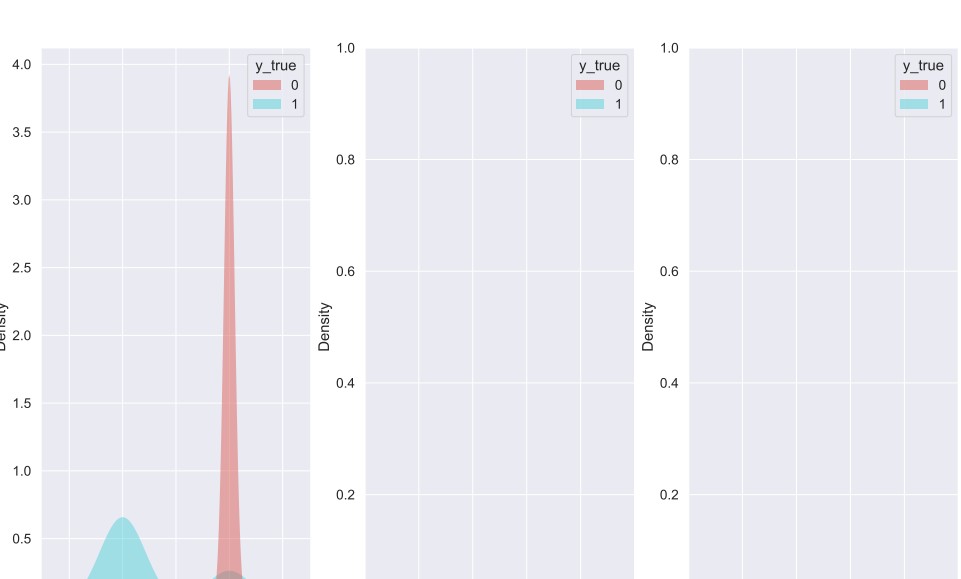

**Figure 10.** Histogram of binary activations generated employing KDE for neurons in hidden layer one in DL predictor for BCWD dataset. Each plot shows the activation density function per neuron discriminated by class (class 0 is shown in red, and class 1 is shown in blue). Neurons h_1_1 and h_1_2 show an empty plot because they have zero variance and present the same activation value for all the samples, thus are not class-discriminatory.

### 6.3.3. From Neurons to Decisions: Activation Pattern

As was discussed above, binary activations in the hidden layers generate binary patterns representing the input examples transformed by each hidden layer. From those binary patterns, we can filter out those neurons with zero variance because they do not apport discriminative information. With the filtered set of binary patterns, we can identify the most frequently active neurons and their activation values per class, generating an *activation pattern*.

Figure 11 (automatically generated) shows the activation pattern for class 0 in the banknote dataset. The activation pattern is highlighted in red and involves neurons: h_1_0 with activation value $-1$, h_2_0 with activation value 1, and the output neuron y_0_1 with activation value 1. The activation pattern reveals layer by layer the most likely decisions the network is making to reach the final decision.

Figure 12 describes the activation pattern for class 1 in the banknote dataset. The activation pattern is highlighted in red and involves neurons: h_1_1 with activation value 1, h_2_0 with activation value -1, and the output neuron y_1_1 with activation value 1. The activation pattern reveals layer by layer the most likely decisions the network makes to reach the final decision. Comparing the activation pattern for class 0 and class 1, we can gain insight into the internal decision process carried out by the binary DL predictor. For example, we know that decision relies on neurons h_1_0 and h_1_1 for the first hidden layer. For the second hidden layer, the decision is taken by neuron h_2_0, which, acting as a switch, selects the output neuron to activate.

Finally, we can infer that the activation pattern is a valuable explainable artifact that complements the rule sets and provides additional insight into the internal decision process carried out by the binary DL predictor.

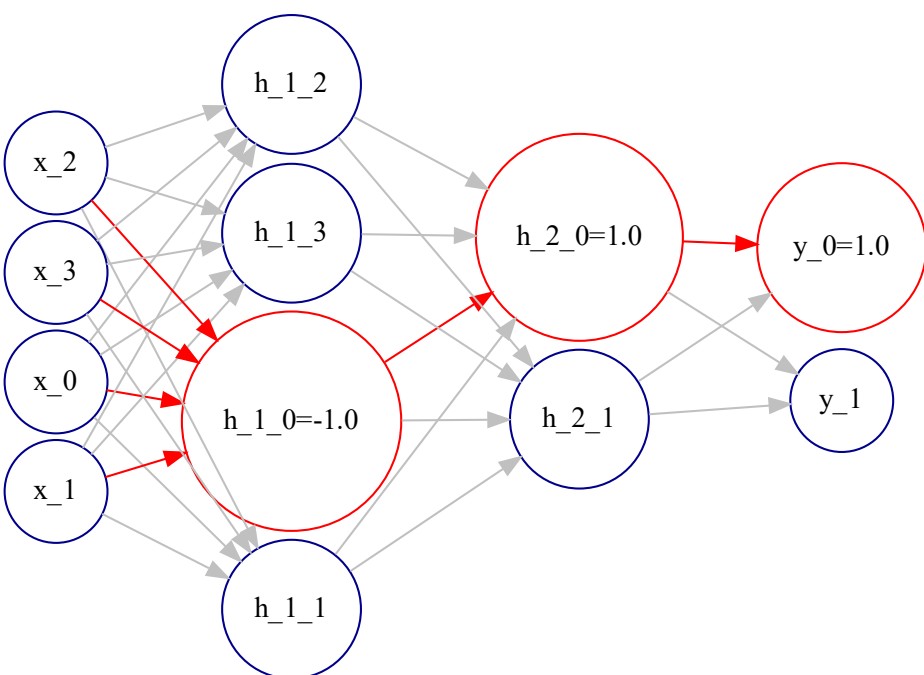

**Figure 11.** Banknote dataset activation pattern (automatically generated) for class 0 is highlighted in red over the DL predictor architecture. In addition, the most frequent activation values are shown for neurons in the activation pattern.

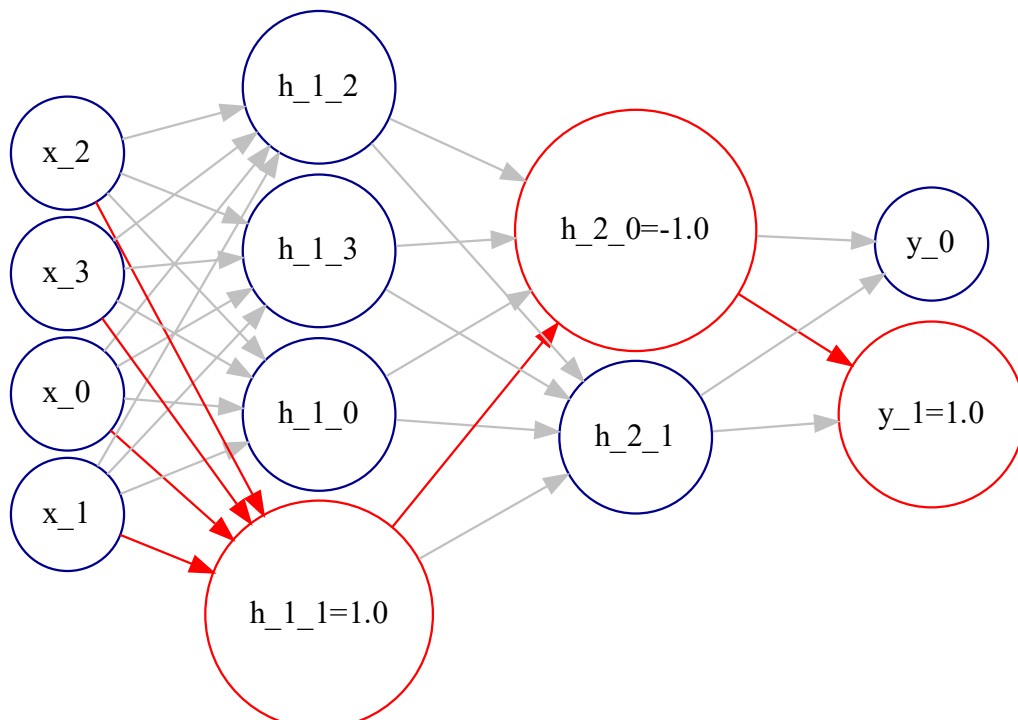

**Figure 12.** The activation pattern (automatically generated) for class 1 is highlighted in red over the DL predictor architecture on the Banknote dataset. In addition, the most frequent activation values are shown for neurons in the activation pattern.

### 6.3.4. Rationales behind Rule Induction

This work employs two methods to induce the feature rules: explainable layers (ExpL) for binary classification tasks (SC1) and decision trees for multiclass classification (SC2).

In scenario SC1, an ExpL is added between the input layer and the first hidden layer. After a fine-tuning training with the rest of the layers frozen, the weights of ExpL codify a threshold value ($v_i$) per feature ($x_i$) that splits the input space in two, indicating the point of maximum information gain per feature. Using the learned threshold values is possible to induce simple rules per feature ($x_i$) in the form of *if* ($x_i \geq v_i$) *then class* 1; *else class* 0, combining these simple rules into the intermediate rule set the final rule set is obtained [60]. ExpL layers are suitable for binary classification tasks, given their ability to split the space into binary buckets with complementary rules. Figure 13 shows the density function discriminated by class for the feature *variance_of_the_wavelet_transform* (class 0 in blue and class 1 in orange), and the learned threshold value ($v_i$) (shown as a vertical red line) that splits the input space for class 0 and 1.

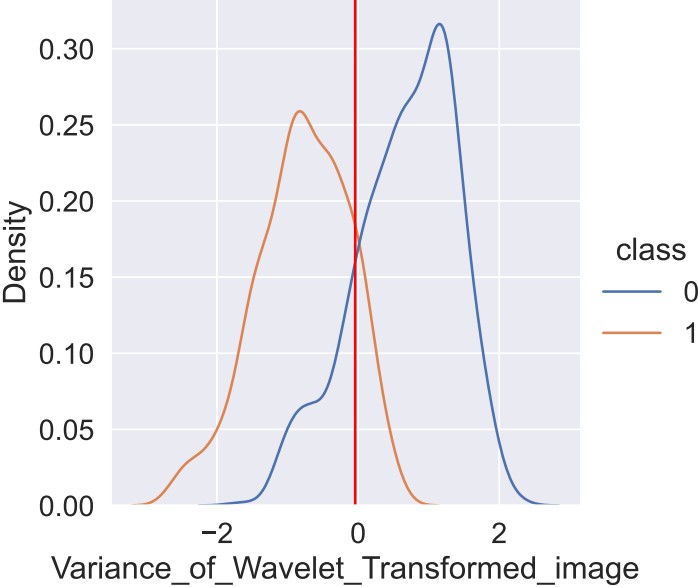

**Figure 13.** the threshold value learned by the explainable layer ExpL (shown as vertical red line) and the density function for the feature *variance_of_the_wavelet_transform* discriminated per class (class 0 is shown in blue, and class 1 is shown in orange).

In the multiclass task case, and given its binary rule induction characteristic, ExpL must employ a series of One-vs-Rest setups. However, one-vs-Rest setups imply a high computational cost due to the increasing number of models to train as the number of classes increases. Equation (9) provides the number of models to train in a One-vs-Rest setup. Applying such a formula to a dataset with three target classes, the required number of models to induce the rule set is 3. However, if we extend the target classes to 10 (Digits dataset), the models to train scale to 45. Additionally, ExpL produces binary splits per class (One-vs-Rest), which are not expressive enough to model the complex decision boundaries of multiclass models, producing low-accurate rule sets in the case of the multiclass task. Due to these reasons, in scenario SC2, we have employed as features' rule inductor the decision trees algorithm (C4.5). Finally, through SC1 and SC2, we have shown that DEXiRE can be applied for both binary and multiclass classification—verifying Hypothesis 4.

$$C_{n_{classes},2} = \frac{n_{classes}!}{2!(n_{classes} - 2!)} \tag{9}$$

### 6.3.5. Beyond Rule Sets

DEXiRE provides characteristics that go beyond other rule-extraction algorithms. In particular:

1. **Tuneable threshold rule set granularity**: DEXiRE provides a fine granularity control over the number of terms in the intermediate and final rule sets. While algorithms such as ECLAIRE control term expansion indirectly through decision tree parameters, DEXiRE provides fine control over the number of terms and the number of clauses through the coverage threshold (Th) parameter. It filters out terms and clauses with low coverage or those conflicting with other rules with higher coverage or confidence values.

2. **Understand per class discriminative neurons**: As discussed above, using binary activations and identifying the activation pattern allow us to obtain clues about the predictor's decision process. It is also possible to identify those neurons discriminating by class and their value and concentrate the analysis on them, removing those neurons with zero variance.

3. **activation pattern supports the knowledge injection**: The activation pattern allows identifying the more frequently activated binary neurons per class in each hidden layer along with its activation value. This information helps to describe the DL estimator's internal decision process, and additionally, the activation pattern can also be used to inject symbolic knowledge within the neural network [69–71]. Symbolic artificial intelligence (AI) relies on high-level knowledge usually expressed as rules, logic predicates, or ontologies [22]. On the other hand, subsymbolic AI is data-driven, as in the case of DL predictors. Nevertheless, subsymbolic and symbolic AI paradigms can be combined following the Hypothesis 2, which states that binary neurons behave as a Boolean logic function. Therefore, first-order logic predicates can be embedded into a Boolean function and then integrated as a binary neuron or layer into the binary DL predictor, with the additional advantage that both symbolic and subsymbolic neurons can co-exist and interact in the same network.

### 6.3.6. Constraints and Limitations

Besides the advantages discussed above, as of today, DEXiRE is still subjected to some constraints (C) and limitations (L):

C: To identify the activation pattern, DEXiRE requires that the output layer has as many neurons as classes and that the class labels are encoded in one hot encoder.

L: Currently, DEXiRE has been tested on the classification task (binary and multiclass classification) with comparable results to the state-of-the-art rule extractors. However, DEXiRE needs to be extended to perform additional tasks, such as rule extraction in the regression task and in the multi-labeling task. These additional capabilities are proposed in future work.

L: To date, DEXiRE can work with a feed-forward neural network (with any number of hidden layers). DEXiRE has not been designed to work with convolutional neural networks (CNN), recurrent neural networks (RNN), or Transformers. However, based on the verified hypotheses, it can also be extended to work with these kinds of DL models.

### 7. Conclusions and Future Work

This paper presents **DEXiRE**, a decompositional rule-extraction algorithm based on hidden layers' binarization, path identification, and layer-by-layer rule induction. DEXiRE has been developed following the Hypotheses 1–4, and it has been tested in two scenarios, binary classification (SC1) and multiclass classification (SC2), with six publicly available datasets.

Attributes and performance of DEXiRE have been compared with ECLAIRE (state-of-the-art for decompositional rule extraction), leading to the conclusion that:

- DEXiRE can be applied to extract rules from DL predictors with **any number of hidden layers**. The layer-by-layer generated rules do not affect nor depend on rules in other layers. The resulting independence reduces the complexity of the intermediate rule sets and eliminates the limitation of being able to process only one or a fixed

predefined number of layers. Indeed, approaches such as FERNN [48] do not scale as they express the rules of a given layer as a function of those from other layers.

- DEXiRE has been tested in binary and multiclass classification scenarios, obtaining performance (accuracy and fidelity) comparable to ECLAIRE.
- Binary neurons with a high degree of polarization in each hidden layer become effective discriminators capable of identifying, through a binary pattern, instances that belong to a particular class. Binary neurons can be expressed as Boolean functions and the relative Boolean intermediate rule sets.
- The activation pattern identifies the most frequently activated binary neurons per class for each layer and filters out irrelevant or redundant neurons. The activation pattern provides additional explainable insights about the DL predictor's internal decision process and reduces the size of the intermediate rule sets and the execution time.

As the feature work, we envision extending DEXiRE to (i) handle regression tasks, (ii) develop a module for convolutional neural networks, and (iii) develop a module for recurrent neural networks.

**Author Contributions:** The contribution to this paper are shared as follows. Conceptualization: D.C., V.C., L.F. and N.M. Methodology: V.C. and D.C. Software: V.C. and N.M.; Validation: V.C., D.C., L.F. and N.M.; State of the Art, G.M., L.F. and Y.M.; Writing—original draft preparation: V.C., D.C. Writing—review and editing: G.M., Y.M., L.F., N.M., J.-P.C. and D.C.; Supervision, Project administration, and Funding acquisition: D.C. and M.S. All authors have read and agreed to the published version of the manuscript.

**Funding:** This work has been par ally supported by the CHISTERA grant CHIST-ERA-19-XAI-005, and by the Swiss National Science Foundation (G.A. 20CH21_195530), the Italian Ministry for Universities and Research, the Luxembourg National Research Fund (G.A. INTER/CHIST/19/14589586), the Scientific and Research Council of Turkey (TÜBİTAK, G.A. 120N680).

**Data Availability Statement:** The datasets employed in this study are open-source and publicly available. They can be found at the links listed below:

- Breast Cancer Wisconsin Diagnosis Dataset: https://www.kaggle.com/datasets/uciml/breast-cancer-wisconsin-data (accessed on 3 November 2022);
- Banknote authentication dataset: https://archive.ics.uci.edu/ml/datasets/banknote+authentication (accessed on 3 November 2022);
- Prima Indians diabetes mellitus dataset: https://www.kaggle.com/datasets/uciml/pima-indians-diabetes-database (accessed on 3 November 2022);
- Digits dataset: https://archive.ics.uci.edu/ml/datasets/Optical+Recognition+of+Handwritten+Digits (accessed on 3 November 2022);
- Iris dataset: https://archive.ics.uci.edu/ml/datasets/iris (accessed on 3 November 2022);
- Wine Quality dataset: https://archive.ics.uci.edu/ml/machine-learning-databases/wine/wine.data (accessed on 3 November 2022);

**Conflicts of Interest:** The authors declare no conflict of interest.

## Appendix A. Datasets Description

*Appendix A.1. BCWD Feature Description*

Table A1 shows the features of the Breast Cancer Wisconsin diagnosis (BCWD). BCWD is composed of 30 features extracted from histopathology images. For every image, ten features were extracted: area, radius, concave points, perimeter, compactness, texture, concavity, smoothness, fractal dimension, and symmetry. For each feature, three values were reported mean, worst value, and standard error (se).

**Table A1.** Breast Cancer Wisconsin Diagnosis (BCWD) Dataset Feature description.

| Feature | Description |
| --- | --- |
| area_se | Area standard error |
| area_worst | Average of three largest area values |
| perimeter_se | Perimeter standard error |
| radius_mean | Average of cell's radius value |
| concave points_worst | Average of three largest concave points in the contour values |
| radius_worst | Average of three largest radius values |
| concave points_mean | Average of concave points in the contour |
| compactness_worst | Average of three largest compactness values |
| concavity_worst | Average of three largest concavity values |
| symmetry_worst | Average of three largest symmetry values |
| smoothness_worst | Average of three largest smoothness values |
| compactness_se | Compactness standard error |
| fractal_dimension_se | Fractal dimension standard error |
| texture_se | Texture standard error |
| fractal_dimension_mean | Fractal dimension mean |
| texture_mean | Texture mean |
| perimeter_mean | Perimeter mean |
| area_mean | Area mean |
| smoothness_mean | Smoothness mean |
| compactness_mean | Compactness mean |
| concavity_mean | Concavity mean |
| symmetry_mean | Symmetry mean |
| radius_se | Radius standard error |
| smoothness_se | Smoothness standard error |
| concavity_se | Concavity standard error |
| concave points_se | Concavity points standard error |
| symmetry_se | Symmetric standard error |
| texture_worst | The average of the largest three texture values |
| perimeter_worst | The average of the largest three texture values |
| fractal_dimension_worst | The average of the largest three fractal dimension values |

*Appendix A.2. Banknote Authentication Dataset*

Table A2 describes the features of the banknote authentication dataset. The Features in this dataset were generated from statistics measures on the wavelet transform from banknotes' images.

**Table A2.** Feature description for the banknote authentication dataset.

| Features | Descriptions |
| --- | --- |
| Feature 1 | variance of Wavelet Transformed image (continuous) |
| Feature 2 | skewness of Wavelet Transformed image (continuous) |
| Feature 3 | kurtosis of Wavelet Transformed image (continuous) |
| Feature 4 | entropy of image (continuous) |

### Appendix A.3. Prima Indians Diabetes Mellitus Classification Dataset

Table A3 describes the features of the Prima Indians diabetes mellitus classification dataset. The Features in this dataset were collected on women older than 21 years old, and its objective is given the features predict if the patient has diabetes or not. Features describe laboratory test results and demography information about the patients.

**Table A3.** Feature description for the Prima Indians diabetes mellitus classification dataset.

| Features | Descriptions |
|---|---|
| Pregnancies | Number of times pregnant. |
| Glucose | Plasma glucose concentration a 2 h in an oral glucose tolerance test. |
| BloodPressure | Diastolic blood pressure (mm Hg). |
| SkinThickness | Triceps skin fold thickness (mm). |
| Insulin | 2-h serum insulin (mu U/mL). |
| BMI | Body mass index (weight in kg/(height in m)$^2$). |
| DiabetesPedigree | Diabetes pedigree function. |
| Age | Age (years) |

### Appendix A.4. Digits Dataset

Table A4 describes the features of the Digits classification dataset. The Features in this dataset represent the intensity of pixels in gray-scale mode.

**Table A4.** Feature description for the digits dataset.

| Features | Descriptions |
|---|---|
| *Pixel_0_0 . . . Pixel_7_7* | Pixel Intensity level |

### Appendix A.5. Iris Dataset

Table A5 describes the features of the Iris dataset. The Features in this dataset represent flowers' sepal and petal length and width. This dataset aims to give the features that classify a flower into three classes (Setosa, Virginica, Versicolor).

**Table A5.** Feature description for the Iris dataset.

| Features | Descriptions |
|---|---|
| Feature 1 | sepal length in centimeters |
| Feature 2 | sepal width in centimeters |
| Feature 3 | petal length in centimeters |
| Feature 4 | petal width in centimeters |

### Appendix A.6. Wine Quality Dataset

The wine quality dataset employs several features that influence the wine quality and taste. Features are used to classify wine into three classes: high, medium, and low quality. Table A6 describes each feature employed in this dataset.

**Table A6.** Feature description for the wine quality dataset.

| Features | Descriptions |
|---|---|
| Feature 1 | Alcohol |
| Feature 2 | Malic acid |
| Feature 3 | Ash |
| Feature 4 | Alkalinity of ash |
| Feature 5 | Magnesium |
| Feature 6 | Total phenols |
| Feature 7 | Flavanoids |
| Feature 8 | Nonflavanoid phenols |
| Feature 9 | Proanthocyanins |
| Feature 10 | Color intensity |
| Feature 11 | Hue |
| Feature 12 | OD280/OD315 of diluted wines |
| Feature 13 | Proline |

## Appendix B. Rule Generated

The Appendix B presents examples of the rules generated per dataset and method (ECLAIRE and DEXiRE).

*Appendix B.1. Breast Cancer Dataset*

Table A7 shows the intermediate rule sets generated by DEXiRE for the BCWD dataset.

**Table A7.** Intermediate rule set, for the BCWD dataset generated by DEXiRE.

| Layer | Rules |
|---|---|
| output layer<br>output layer | *IF $h_{3,0}$ THEN Benign*<br>*IF $h_{3,1}$ THEN Malign* |
| hidden layer 1<br>hidden layer 1 | *IF $h_{1,0}$ THEN Benign*<br>*IF $\neg h_{1,0}$ THEN Malign* |
| hidden layer 0<br>hidden layer 0 | *IF $h_{0,0} \vee \neg h_{0,8}$ THEN Benign*<br>*IF $\neg h_{0,0} \wedge h_{0,8}$ THEN Malign* |

In Table A8 is shown the final rule set, for BCWD dataset inducted by DEXiRE on this dataset.

**Table A8.** Final rule set, for the BCWD dataset generated by DEXiRE.

| Rules |
|---|
| *IF $[(concave\_points\_worst > 0.388)] \vee [(perimeter\_mean > 0.4003)]$ THEN Malign*<br>*IF $[(concave\_points\_worst \leq 0.388) \wedge (perimeter\_mean \leq 0.4003)]$ THEN Benign* |

Table A9 shows the final rule set for the BCWD dataset inducted by ECLAIRE.

**Table A9.** Final rule set, for the BCWD dataset generated by ECLAIRE.

| Rules |
|---|
| *IF $[(perimeter\_worst \leq 0.2355)] \vee [(concave\_points\_worst \leq 0.388)]$ THEN Benign*<br>*IF $[(concave\_points\_worst > 0.388)] \vee [(perimeter\_worst > 0.2355)]$ THEN Malign* |

*Appendix B.2. Banknote Authentication Dataset*

Table A10 shows the intermediate rule set generated by DEXiRE for the banknote authentication dataset.

**Table A10.** DEXiRE Intermediate rule set for the banknote authentication dataset.

| Layer | Rules |
|---|---|
| output layer | $IF\ h_{1,0}\ THEN\ 0$ |
| output layer | $IF\ h_{1,2}\ THEN\ 1$ |
| Hidden layer 0 | $IF\ h_{0,0} \wedge h_{0,1}\ THEN\ 0$ |
| Hidden layer 0 | $IF\ \neg h_{0,0} \wedge h_{0,2}\ THEN\ 1$ |

In Table A11 is shown the final rule set inducted by DEXiRE for the banknote authentication dataset.

**Table A11.** Final rule set, for the banknote authentication dataset generated by DEXiRE.

| Rule | | | |
|---|---|---|---|
| $IF[(Kurtosis\_of\_Wavelet\_Transformed\_image$ | $>$ | $0.3402) \wedge$ | |
| $(Skewness\_of\_Wavelet\_Transformed\_image$ | $>$ | $-0.6451)] \vee$ | |
| $[(Skewness\_of\_Wavelet\_Transformed\_image$ | $>$ | $0.486) \wedge$ | |
| $(Variance\_of\_Wavelet\_Transformed\_image$ | $>$ | $-1.0579)] \vee$ | |
| $[(Skewness\_of\_Wavelet\_Transformed\_image$ | $>$ | $0.328) \wedge$ | |
| $(Variance\_of\_Wavelet\_Transformed\_image$ | $>$ | $0.1144)] \vee$ | |
| $[(Kurtosis\_of\_Wavelet\_Transformed\_image$ | $>$ | $-0.7389) \wedge$ | |
| $(Variance\_of\_Wavelet\_Transformed\_image > 0.1144)]\ THEN\ 0$ | | | |
| $IF[(Kurtosis\_of\_Wavelet\_Transformed\_image$ | $\leq$ | $-0.7389) \wedge$ | |
| $(Skewness\_of\_Wavelet\_Transformed\_image$ | $\leq$ | $0.328)] \vee$ | |
| $[(Variance\_of\_Wavelet\_Transformed\_image$ | $\leq$ | $-1.0579)] \vee$ | |
| $[(Skewness\_of\_Wavelet\_Transformed\_image$ | $\leq$ | $-0.6451) \wedge$ | |
| $(Variance\_of\_Wavelet\_Transformed\_image$ | $\leq$ | $0.1144)] \vee$ | |
| $[(Kurtosis\_of\_Wavelet\_Transformed\_image$ | $\leq$ | $0.3402) \wedge$ | |
| $(Skewness\_of\_Wavelet\_Transformed\_image$ | $\leq$ | $0.486) \wedge$ | |
| $(Variance\_of\_Wavelet\_Transformed\_image \leq 0.1144)]\ THEN\ 1$ | | | |

Table A12 shows the rules obtained from ECLAIRE for the banknote authentication dataset.

**Table A12.** Final rule set for the Banknote authentication dataset generated by ECLAIRE.

| Rules | | | |
|---|---|---|---|
| $IF[(Kurtosis\_of\_Wavelet\_Transformed\_image$ | $>$ | $1.2494) \wedge$ | |
| $(Variance\_of\_Wavelet\_Transformed\_image$ | $>$ | $-0.7811)] \vee$ | |
| $[(Variance\_of\_Wavelet\_Transformed\_image$ | $>$ | $0.1144)] \vee$ | |
| $[(Skewness\_of\_Wavelet\_Transformed\_image$ | $>$ | $0.5262) \wedge$ | |
| $(Variance\_of\_Wavelet\_Transformed\_image > -1.0579)]THEN\ 0$ | | | |
| $IF[(Kurtosis\_of\_Wavelet\_Transformed\_image$ | $\leq$ | $1.2494) \wedge$ | |
| $(Skewness\_of\_Wavelet\_Transformed\_image$ | $\leq$ | $0.5262) \wedge$ | |
| $(Variance\_of\_Wavelet\_Transformed\_image$ | $\leq$ | $0.1144)] \vee$ | |
| $[(Variance\_of\_Wavelet\_Transformed\_image$ | $\leq$ | $-1.0579)] \vee$ | |
| $[(Skewness\_of\_Wavelet\_Transformed\_image$ | $\leq$ | $0.5262) \wedge$ | |
| $(Variance\_of\_Wavelet\_Transformed\_image \leq -0.7811)]\ THEN\ 1$ | | | |

*Appendix B.3. Prima Indians Diabetes Dataset*

In Table A13 are shown the intermediate rule sets generated by DEXiRE for the Prima Indians dataset.

**Table A13.** DEXiRE Intermediate rule sets for the Prima Indians dataset.

| Layer | Rules |
|---|---|
| output layer | *IF* $h_{1,0}$ *THEN* 0 |
| output layer | *IF* $h_{1,2}$ *THEN* 1 |
| Hidden layer 0 | *IF* $h_{0,0} \wedge h_{0,1}$ *THEN* 0 |
| Hidden layer 0 | *IF* $\neg h_{0,0} \wedge h_{0,2}$ *THEN* 1 |

In Table A14 is shown the final rule set for Prima Indians diabetes dataset, inducted by DEXiRE on this dataset.

**Table A14.** Final rule set, for the Prima Indians diabetes dataset, generated by DEXiRE.

| Rule |
|---|
| *IF* $[(BMI > 0.8576) \wedge (Glucose \leq -0.341)] \vee [(BMI \leq -0.2212) \wedge (Glucose > 0.2067) \wedge (Pregnancies \leq -0.1025)] \vee [(BMI \leq 0.8576) \wedge (Glucose \leq 0.2067)]$ *THEN* 0 |
| *IF* $[(BMI \leq -0.2212) \wedge (Glucose > 0.2067) \wedge (Pregnancies > -0.1025)] \vee [(BMI > 0.8576) \wedge (Glucose \leq 0.2067) \wedge (Glucose > -0.341)] \vee [(BMI > -0.2212) \wedge (Glucose > 0.2067)]$ *THEN* 1 |

Table A15 shows the final rule set generated by ECLAIRE for Prima Indian diabetes dataset.

**Table A15.** Final rule set for the Prima Diabetes dataset generated by ECLAIRE.

| Rules |
|---|
| *IF* $[(Glucose > 0.6918)] \vee [(Pregnancies > 0.343) \wedge (SkinThickness > 0.7191)] \vee [(BMI > 0.7878)(Glucose > -0.028)] \vee [(BMI > 0.4832) \wedge (Glucose > -0.1219)] \vee [(Glucose > 0.7857)] \vee [(BMI > -0.2021) \wedge (Glucose > 0.6605)]$ *THEN* 1 |
| *IF* $[(BMI \leq 0.0898)] \vee [(Glucose \leq 0.0346) \wedge (SkinThickness \leq 1.0327)] \vee [(Glucose \leq -0.1532)] \vee [(Glucose \leq -0.1219) \wedge (Pregnancies \leq 0.6399)] \vee [(BMI \leq 0.3309) \wedge (Glucose \leq 0.1598)] \vee [(Glucose \leq 0.0659) AND (SkinThickness \leq 0.5936)] \vee [(BMI \leq -0.2656) AND (Glucose \leq 0.3476) \wedge (SkinThickness \leq 0.5936)]$ *THEN* 0 |

*Appendix B.4. Digits Dataset*

Table A16 shows the final rule set for the digits dataset generate by DEXiRE.

**Table A16.** Final rule set for the digits dataset generated by DEXiRE.

| Rules |
|---|
| *IF* $[(pixel\_4\_4 > -1.7367)] \vee [(pixel\_2\_3 \leq 0.8632) \wedge (pixel\_2\_5 > -1.26) \wedge (pixel\_3\_2 \leq -0.4994) \wedge (pixel\_4\_4 > -1.7367) \wedge (pixel\_6\_5 > -1.4526)]$ *THEN* 3 |
| *IF* $[(pixel\_2\_5 \leq -1.26) \wedge (pixel\_4\_4 > -1.7367) \wedge (pixel\_5\_2 > 0.1711) \wedge (pixel\_6\_4 > 0.1101)] \vee [(pixel\_1\_5 \leq 0.1362) \wedge (pixel\_2\_5 > -1.26) \wedge (pixel\_3\_2 > -0.3378) \wedge (pixel\_4\_1 > 0.1898) \wedge (pixel\_4\_4 > -1.7367)]$ *THEN* 4 |

**Table A16.** *Cont.*

| Rules |
| --- |
| *IF*[(*pixel_2_5* > −1.26) ∧ (*pixel_3_2* ≤ −0.3378) ∧ (*pixel_4_1* > 0.1898) ∧ (*pixel_4_4* > −1.7367)] ∨ [(*pixel_2_3* ≤ 0.8632) ∧ (*pixel_2_5* > −1.26) ∧ (*pixel_4_1* ≤ 0.1898) ∧ (*pixel_4_4* > −1.7367) ∧ (*pixel_7_4* ≤ −0.975)] ∨ [(*pixel_2_3* ≤ 0.8632) ∧ (*pixel_3_2* ≤ −0.4994) ∧ (*pixel_4_1* ≤ 0.1898) ∧ (*pixel_4_4* > −1.7367) ∧ (*pixel_6_5* ≤ −1.4526)] ∨ [(*pixel_2_3* ≤ 0.8632) ∧ (*pixel_2_5* > −1.26) ∧ (*pixel_3_2* ≤ 0.3083) ∧ (*pixel_3_2* > −0.4994) ∧ (*pixel_4_4* > −1.7367) ∧ (*pixel_7_4* ≤ −0.975)]*THEN*7 |
| *IF*[(*pixel_2_3* > 0.8632) ∧ (*pixel_2_5* > −1.26) ∧ (*pixel_4_1* ≤ 0.1898) ∧ (*pixel_4_4* ≤ −0.0508)] ∨ [(*pixel_4_1* ≤ −0.6724) ∧ (*pixel_4_4* ≤ −1.7367)] ∨ [(*pixel_2_3* ≤ 0.8632) ∧ (*pixel_2_5* > −1.26) ∧ (*pixel_3_2* > −0.4994) ∧ (*pixel_4_1* ≤ 0.1898) ∧ (*pixel_4_4* > −1.7367) ∧ (*pixel_4_6* > −0.8227) ∧ (*pixel_7_4* > −0.975)]*THEN*9 |
| *IF*[(*pixel_1_5* > 0.1362) ∧ (*pixel_2_5* > −1.26) ∧ (*pixel_3_2* > −0.3378) ∧ (*pixel_4_1* > 0.1898) ∧ (*pixel_4_4* > −1.7367)] ∨ [(*pixel_2_3* > 0.8632) ∧ (*pixel_2_4* > 0.794) ∧ (*pixel_2_5* > −1.26) ∧ (*pixel_4_1* ≤ 0.1898) ∧ (*pixel_4_4* > −0.0508)] ∨ [(*pixel_0_5* ≤ −0.8441) ∧ (*pixel_2_5* ≤ −1.26) ∧ (*pixel_3_3* > 0.5405) ∧ (*pixel_4_6* ≤ −0.8227) ∧ (*pixel_5_2* ≤ 0.1711)]*THEN*1 |
| *IF*[(*pixel_2_3* > 0.8632) ∧ (*pixel_2_4* ≤ 0.794) ∧ (*pixel_2_5* > −1.26) ∧ (*pixel_4_1* ≤ 0.1898) ∧ (*pixel_4_4* > −0.0508)] ∨ [(*pixel_2_5* > −1.26) ∧ (*pixel_4_1* ≤ 0.1898) ∧ (*pixel_4_4* > −1.7367) ∧ (*pixel_5_2* > 0.3241) ∧ (*pixel_5_5* > −0.3928) ∧ (*pixel_6_5* > −1.4526)] ∨ [(*pixel_2_3* ≤ 0.8632) ∧ (*pixel_2_5* > −1.26) ∧ (*pixel_3_2* > −0.4994) ∧ (*pixel_4_1* ≤ 0.1898) ∧ (*pixel_4_4* > −1.7367) ∧ (*pixel_4_6* ≤ −0.8227) ∧ (*pixel_7_4* > −0.975)]*THEN*8 |
| *IF*[(*pixel_0_5* ≤ −0.8441) ∧ (*pixel_2_5* ≤ −1.26) ∧ (*pixel_3_3* ≤ 0.5405) ∧ (*pixel_4_6* ≤ −0.8227) ∧ (*pixel_5_2* ≤ 0.1711)] ∨ [(*pixel_2_3* ≤ 0.8632) ∧ (*pixel_3_2* ≤ −0.4994) ∧ (*pixel_4_4* > −1.7367) ∧ (*pixel_5_5* ≤ −0.3928) ∧ (*pixel_6_5* > −1.4526)]*THEN*24 |
| *IF*[(*pixel_4_1* > −0.6724) ∧ (*pixel_4_4* ≤ −1.7367)]*THEN*0 |
| *IF*[(*pixel_0_5* > −0.8441) ∧ (*pixel_2_5* ≤ −1.26) ∧ (*pixel_5_2* ≤ 0.1711)]*THEN*5 |
| *IF*[(*pixel_2_5* ≤ −1.26) ∧ (*pixel_4_4* > −1.7367) ∧ (*pixel_5_2* > 0.1711) ∧ (*pixel_6_4* ≤ 0.1101)]*THEN*6 |

Table A17 shows the final rule set for the Digits dataset inducted by ECLAIRE.

**Table A17.** Final rule set for the digits dataset generated by ECLAIRE.

| Rules |
| --- |
| *IF*[(*pixel_2_2* ≤ −1.2134) ∧ (*pixel_5_3* ≤ −0.8119)] ∨ [(*pixel_3_2* ≤ −0.6609) ∧ (*pixel_3_6* ≤ −0.6289) ∧ (*pixel_5_2* ≤ 0.1711) ∧ (*pixel_6_3* ≤ 0.0882)] ∨ [(*pixel_2_3* ≤ 0.1736) ∧ (*pixel_3_2* ≤ −0.984) ∧ (*pixel_5_5* > −0.3928) ∧ (*pixel_6_3* ≤ 0.6623)]*THEN* 3 |
| *IF*[(*pixel_4_6* ≤ −0.8227) ∧ (*pixel_5_2* > −0.5938) ∧ (*pixel_6_3* ≤ 0.4709)] ∨ [(*pixel_2_5* > 0.1927) ∧ (*pixel_3_3* > −0.4797) ∧ (*pixel_4_6* ≤ −0.8227) ∧ (*pixel_5_2* > −0.5938)] ∨ [(*pixel_4_6* ≤ −0.8227) ∧ (*pixel_5_2* > −0.5938) ∧ (*pixel_5_3* ≤ 0.2751) ∧ (*pixel_6_3* ≤ 0.4709)]*THEN* 8 |
| *IF*[(*pixel_2_2* > 0.1928) ∧ (*pixel_2_5* ≤ −1.0986) ∧ (*pixel_5_2* ≤ 0.1711) ∧ (*pixel_7_5* ≤ 0.2095)] ∨ [(*pixel_0_6* > −0.4097) ∧ (*pixel_2_5* ≤ −1.0986)]*THEN* 5 |

**Table A17.** *Cont.*

| Rules |
| --- |
| $IF[(pixel\_3\_2 \le 0.3083) \wedge (pixel\_4\_4 > 0.4549) \wedge (pixel\_7\_4 \le -0.975)] \vee$ $[(pixel\_3\_2 \le 0.3083) \wedge (pixel\_4\_4 > 0.4549) \wedge (pixel\_7\_4 \le -1.1777)]THEN 7$ |
| $IF[(pixel\_3\_6 > -0.6289) \wedge (pixel\_5\_2 \le -0.4409) \wedge (pixel\_5\_4 \le -0.5868)] \vee$ $[(pixel\_3\_2 > -1.4686) \wedge (pixel\_3\_5 > 0.0764) \wedge (pixel\_5\_2 \le 0.0181) \wedge$ $(pixel\_5\_4 \le -1.0662)] \vee [(pixel\_3\_5 > 0.9281) \wedge (pixel\_4\_2 \le -0.2637) \wedge$ $(pixel\_5\_4 \le -0.5868)] \vee [(pixel\_3\_5 > 0.9281) \wedge (pixel\_5\_2 \le -0.4409) \wedge$ $(pixel\_5\_4 \le -0.5868)]THEN 9$ |
| $IF[(pixel\_4\_1 > 0.4772) \wedge (pixel\_7\_2 \le -1.0894)] \vee [(pixel\_4\_1 > 1.6269)] \vee$ $[(pixel\_0\_2 \le -0.8846) \wedge (pixel\_1\_5 \le -0.8553) \wedge (pixel\_4\_1 > 0.1898) \wedge$ $(pixel\_5\_4 > -0.1074)] \vee [(pixel\_0\_2 \le -0.8846) \wedge (pixel\_1\_5 \le -0.5248) \wedge$ $(pixel\_5\_1 > 1.146)]THEN 4$ |
| $IF[(pixel\_1\_2 \le -1.7311) \wedge (pixel\_2\_3 \le 1.0355) \wedge (pixel\_3\_3 > 0.2004)] \vee$ $[(pixel\_1\_2 \le -1.7311) \wedge (pixel\_2\_3 \le 1.0355) \wedge (pixel\_2\_4 > 1.2799)] \vee$ $[(pixel\_1\_2 \le 0.483) \wedge (pixel\_2\_3 > 1.0355) \wedge (pixel\_2\_4 > 0.47) \wedge (pixel\_4\_6 \le$ $-0.8227)] \vee [(pixel\_1\_2 \le -0.0705) \wedge (pixel\_2\_3 > 1.0355) \wedge (pixel\_2\_4 >$ $0.956)] \vee [(pixel\_2\_3 > 1.0355) \wedge (pixel\_7\_7 > 0.8795)]THEN 1$ |
| $IF[(pixel\_2\_3 \le 1.2079) \wedge (pixel\_7\_7 > -0.196)] \vee [(pixel\_4\_5 \le -1.4899) \wedge$ $(pixel\_6\_6 > -0.7574) \wedge (pixel\_7\_7 \le -0.196)] \vee [(pixel\_2\_3 \le 0.6908) \wedge$ $(pixel\_7\_7 > -0.196)] \vee [(pixel\_3\_2 \le -0.984) \wedge (pixel\_4\_3 \le 0.3077) \wedge$ $(pixel\_4\_5 \le -0.2972) \wedge (pixel\_6\_3 > 0.0882)]THEN 2$ |
| $IF[(pixel\_3\_4 \le -0.9637) \wedge (pixel\_4\_4 \le -1.7367)]THEN 0$ |
| $IF[(pixel\_2\_5 \le -1.26) \wedge (pixel\_5\_2 > 0.1711) \wedge (pixel\_6\_1 \le 0.7421) \wedge$ $(pixel\_6\_4 \le -0.0785)]THEN 6$ |

*Appendix B.5. Iris Dataset*

Table A18 shows the final rule set for the Iris dataset inducted by DEXiRE.

**Table A18.** Final rule set for the Iris dataset generated by DEXiRE.

| Rules |
| --- |
| $IF [(petal\_length\_(cm) > 0.4713)] THEN 2$ |
| $IF [(petal\_length\_(cm) \le 0.4713)AND(petal\_length\_(cm) > -1.0618)] THEN 1$ |
| $IF [(petal\_length\_(cm) \le -1.0618)] THEN 0$ |

Table A19 shows the final rule set for the Iris dataset inducted by ECLAIRE.

**Table A19.** Final rule set for the Iris dataset generated by ECLAIRE.

| Rules |
| --- |
| $IF [(petal\_width\_(cm) \le 0.6518) \wedge (sepal\_width\_(cm) \le -0.1082)] THEN 1$ |
| $IF [(petal\_width\_(cm) > 0.3884)] THEN 2$ |
| $IF [(petal\_width\_(cm) \le -0.7966)] THEN 0$ |

*Appendix B.6. Wine Quality Dataset*

**Table A20.** Final rule set for the wine quality dataset generated by DEXiRE.

| Rules |
| --- |
| $IF [(hue \le -0.2959) \wedge (proline \le 0.6054)] THEN 2$ |
| $IF [(hue > -0.2959) \wedge (proline \le 0.6054)] THEN 1$ |
| $IF [(proline > 0.6054)] THEN 0$ |

**Table A21.** Final rule set for the wine quality dataset generated by ECLAIRE.

| Rules |
|---|
| *IF* $[(od280/od315\_of\_diluted\_wines \leq -0.7086)]$ *THEN* 2 |
| *IF* $[(proline > 0.2806)] \lor [(proline > 0.8538)]$ *THEN* 0 |
| *IF* $[(color\_intensity \leq -0.4837)] \lor [(color\_intensity \leq -0.1636) \land (magnesium \leq -0.6138)]$ *THEN* 1 |

## Appendix C. Model Analysis

*Appendix C.1. Binary Classification*

Appendix C.1.1. Banknote Dataset

In Figure A1 is shown the histogram of binary activation in hidden layer 0. In several neurons there is a complementary activation between classes 0 (identified with color red) and 1 (identified with color blue).

Histogram of activations BNN layer 0

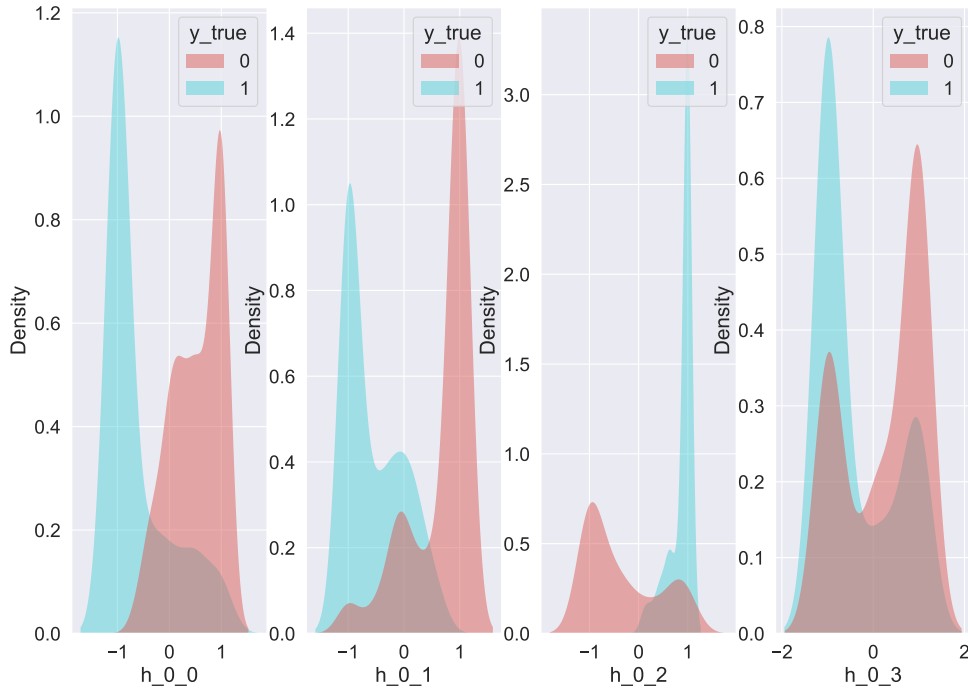

**Figure A1.** Histogram of binary activations for neurons in hidden layer zero in DL predictor for banknote authentication dataset. Each plot shows the activation density function per neuron discriminated by class (class 0 is shown in red, and class 1 is shown in blue).

Appendix C.1.2. Prima Indians Diabetes

Figure A2 shows the histogram of binary activation in hidden layer 0. In several neurons there is clear a strong complementary activation between classes 0 (identified with color red) and 1 (identified with color blue). This dataset exhibits greater complexity in its activation pattern than the previous one, producing more complex rule sets.

Histogram of activations BNN layer 0

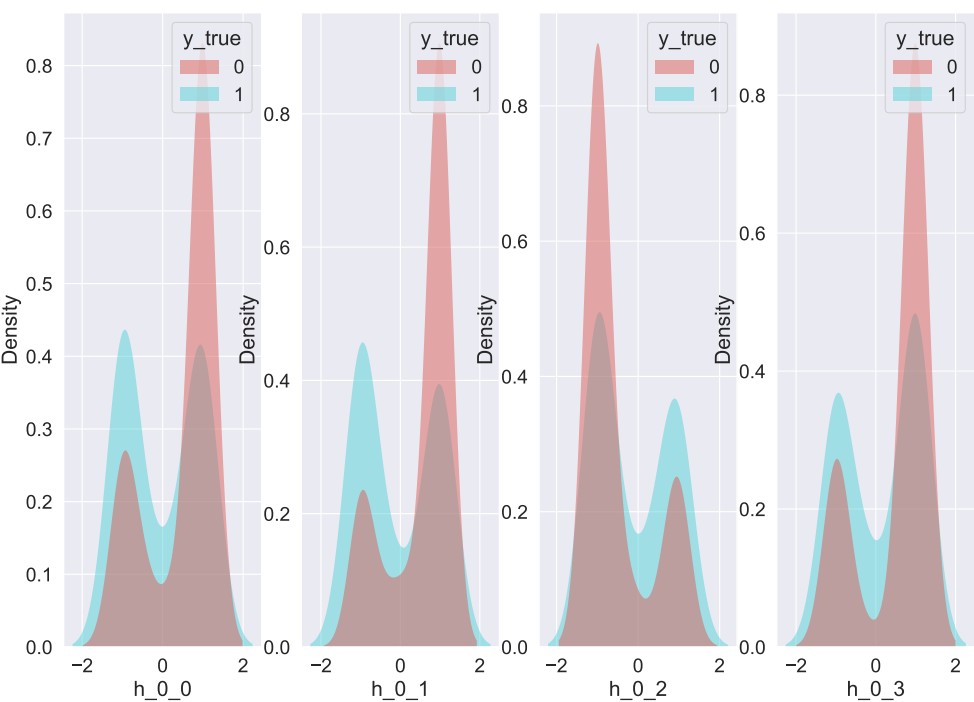

**Figure A2.** Histogram of binary activations for neurons in hidden layer zero in DL predictor for Prima Indians diabetes dataset. Each plot shows the activation density function per neuron discriminated by class (class 0 is shown in red, and class 1 is shown in blue).

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
