# Peer review of "A DEXiRE for Extracting Propositional Rules from Neural Networks via Binarization"

_electronics, doi:10.3390/electronics11244171_

Round 1
Reviewer 1 Report
1- The abstract is too long the default abstract is between 150-250 words but in this manuscript is more than 300 words.
2- figures 1-5 missed the cited reference
3- Algorithm 1 is not clear or not understandable.
4- To improve the result, the authors shall compare it with other related work and present the result based on the paper result.
5- The references format needed to justify
Author Response
We would like to thank the editors and the reviewers for all the detailed and useful feedback we have received. Thanks to that, the manuscript has notably improved. Below are detailed answers to the reviewers' notes.
Once again, thank you for your effort.
Reviewer #1:
[R1.1]: The abstract is too long the default abstract is between 150-250 words, but in this manuscript is more than 300 words.
[A1.1]: The abstract has been shortened to comply with the requirements. The word count tool of MS Word and Grammarly report 248 for current version (see below).
Background: Despite the advancement in eXplainable Artificial Intelligence, the explanations provided by model-agnostic predictors still call for improvements (i.e., lack of accurate descriptions of predictors' behaviors).
Contribution: We present a tool for Deep Explanations and Rule Extraction (DEXiRE) to approximate rules for Deep Learning models with any number of hidden layers.
Methodology: DEXiRE proposes the binarization of neural networks to induce boolean functions in the hidden layers, generating as many intermediate rule sets. A rule set is inducted between the first hidden layer and the input layer. Finally, the complete rule set is obtained using inverse substitution on intermediate rule sets and first-layer rules. Statistical tests and satisfiability algorithms reduce the final rule set's size and complexity (filtering redundant, inconsistent, and non-frequent rules). DEXiRE has been tested in binary and multiclass classifications, with six datasets having different structures and models.
Results: The performance is consistent (in terms of accuracy, fidelity, and rule length) with respect to the state-of-the-art rule extractors (i.e., ECLAIRE). Moreover, compared with ECLAIRE, DEXiRE has generated shorter rules (i.e., up to 74\% fewer terms) and has shortened the execution time (improving up to $197\%$ in the best-case scenario).
Conclusions: DEXiRE can be applied for binary and multiclass classification of deep learning predictors with any number of hidden layers. Moreover, DEXiRE can identify the activation pattern per class and use it to reduce the search space for rule extractors (pruning irrelevant/redundant neurons) --- shorter rules and execution times with respect to the ECLAIRE.
[R1.2]: figures 1-5 missed the cited reference
[A1.2]: Figures 1-5 are original and generated by the authors in the context of this paper. In particular, Figure 1 has been generated based on our study of state-of-the-art. Figure 2 shows the pipeline and step-by-step execution of the DEXiRE algorithm originally proposed in this paper. Figures 3 and 4 have been generated (automatically) from the actual DL predictor and DEXiRE process. Figure 5 is an illustrative figure that shows the substitution process produced by our algorithm. Moreover, we have checked that all the figures have been properly referred to in the manuscript.
[R1.3]: Algorithm 1 is not clear or not understandable.
[A1.3]: Algorithm 1 is expressed in pseudocode, and it actualizes the pipeline schematically depicted in Figure 2 (which has been moved to the beginning of the section for the sake of clarity). Moreover, a short description in Section 3.2 of the process/algorithm has been introduced at the beginning of the section. Finally, additional explanatory comments have been added to Algorithm 1 to connect with the step-by-step explanation presented in the text.
[R1.4]: To improve the result, the authors shall compare it with other related work and present the result based on the paper result.
[A1.4]: We used the state-of-the-art tool (ECLAIRE) acknowledging the source and the original paper, and we tested it on the same datasets (on 6 datasets and 2 scenarios) with the same machine; comparative results between ECLAIRE and DEXiRE can be found in section 5 (code and repositories will be released with the camera ready).
[R1.5]: The references format needed to justify
[A1.5]: We are afraid to have not clearly understood this point. However, if it refers to the format of the references, it will be rectified during the production of the camera ready (as it will be required).
Reviewer 2 Report
I am delighted to have the opportunity to review the work of the authors. The authors studied ratio rule extraction in neural networks binarization binary to improve the limitations of XAI. This paper has been structured in a solid and systematic way. The author guided the overall flow well and emphasized the heart of the study. However, if the following improvements are made, readers will read this manuscript more interestingly.
1 I want researcher to express more clearly what they ultimately want to emphasize through this paper. In the abstract, the authors pointed to the accurate description of behaviors of model-agnostic predictors. Various indicators were presented throughout the manuscript to emphasize the superiority of the proposed methodology. What exactly should readers focus on to understand this paper? accuracy? Execution time? It would be better to revise the expression of the abstract or to remind the flow of the manuscript what the main purpose of this study is.
2 The authors proposed four hypotheses. Throughout the manuscript, the purpose of the study, performance indicators, and hypotheses are scattered. I recommend that authors plot the relationship between the study's eventual goal, performance indicators, and hypothesis testing.
3 The manuscript contains content that is not directly related to the core of the study. I know the greed and sincerity of the researchers, but in order to express the main findings more clearly, I propose to remove the most indirect content first.
Author Response
We would like to thank the editors and the reviewers for all the detailed and useful feedback we have received. Thanks to that, the manuscript has notably improved. Below are detailed answers to the reviewers' notes.
Once again, thank you for your effort.
Reviewer #2:
[R2.1]: I am delighted to have the opportunity to review the work of the authors. The authors studied ratio rule extraction in neural networks binarization binary to improve the limitations of XAI. This paper has been structured in a solid and systematic way. The author guided the overall flow well and emphasized the heart of the study. However, if the following improvements are made, readers will read this manuscript more interestingly.
[A2.1]: We thank the reviewer for such words. We are pleased to hear that our effort is appreciated.
[R2.2]: I want researcher to express more clearly what they ultimately want to emphasize through this paper. In the abstract, the authors pointed to the accurate description of behaviors of model-agnostic predictors. Various indicators were presented throughout the manuscript to emphasize the superiority of the proposed methodology. What exactly should readers focus on to understand this paper? accuracy? Execution time? It would be better to revise the expression of the abstract or to remind the flow of the manuscript what the main purpose of this study is.
[A2.2]: The limit of 250 words for the abstract and the requirement to cover background, contribution, methodology, results, and conclusion do not leave much room for extensive presentations. However, we hermetically stated in the contribution that the reader should focus on DEXiRE (a tool for Deep Explanations and Rule Extraction to approximate rules for Deep Learning models with any number of hidden layers). The evaluation metrics measuring the performance have been made more clear in the paragraph results of the abstract are accuracy, fidelity, and rule length.
[R2.3]: The authors proposed four hypotheses. Throughout the manuscript, the purpose of the study, performance indicators, and hypotheses are scattered. I recommend that the authors plot the relationship between the study's eventual goal, performance indicators, and hypothesis testing.
[A2.3]: we have introduced and contextualized a figure (Figure 2) to represent schematically the connections between the hypotheses, the metrics, and the goal.
[R2.4]: The manuscript contains content that is not directly related to the core of the study. I know the greed and sincerity of the researchers, but in order to express the main findings more clearly, I propose to remove the most indirect content first.
[A2.4]: We have put all the underlying concepts and the relevant state-of-the-art that we believe to be functional to understand the contribution and also for a broader audience.
Reviewer 3 Report
The paper is extremely well-written. All the details are clearly explained. The algorithm steps are explained in great detail. The results are impressive. The testing was done on many carefully chosen benchmark data sets. The comparisons were clear. The tables and figures throughout the paper were very helpful in lending to the understanding. The appendices of rules were interesting and added clarification to the workings of the rule extraction. Great Job overall!
I found one minor item in Hypothesis #4 (line 201). I think it needs to be restated. I believe the "it" should be replaced with "then there exists."
I also think that figure 5 comes too late in the manuscript (line 207). I would have liked to see it at the top. I understand the placement, but I did scroll down to look at it when it was first mentioned.
I also saw a symbol that I was not familiar with in line 251 |=
Author Response
We would like to thank the editors and the reviewers for all the detailed and useful feedback we have received. Thanks to that, the manuscript has notably improved. Below are detailed answers to the reviewers' notes.
Once again, thank you for your effort.
Reviewer #3:
[R3.1]: The paper is extremely well-written. All the details are clearly explained. The algorithm steps are explained in great detail. he results are impressive. The testing was done on many carefully chosen benchmark data sets. The comparisons were clear. The tables and figures throughout the paper were very helpful in lending to the understanding. The appendices of rules were interesting and added clarification to the workings of the rule extraction. Great Job overall.
[A3.1]: We thank the reviewer for such words. We are pleased to hear that our effort is appreciated.
[R3.2]: I found one minor item in Hypothesis #4 (line 201). I think it needs to be restated. I believe the "it" should be replaced with "then there exists."
[A3.2]: We have rephrased hypothesis 4 to improve wording, correctness, and clarity.
[R3.3]: I also think that figure 5 comes too late in the manuscript (line 207). I would have liked to see it at the top. I understand the placement, but I did scroll down to look at it when it was first mentioned. I also saw a symbol that I was not familiar with in line 251 |=
[A3.3]: We thank the reviewer for the input. We understand and agree with it. Hence, Figure 5 has been moved to the beginning of the section to make the step-by-step pipeline easier to be understood (now it is Figure 3). Moreover, a short verbose description has been added to introduce/explain the symbol in line 251, which is referred to as “semantically entailed between a logic model M and conclusion y.”